# Online Convex Optimization with Stochastic Constraints

**Hao Yu,   Michael J. Neely,   Xiaohan Wei**
Department of Electrical Engineering, University of Southern California[*]
{yuhao,mjneely,xiaohanw}@usc.edu

## Abstract

This paper considers online convex optimization (OCO) with stochastic constraints, which generalizes Zinkevich's OCO over a known simple fixed set by introducing multiple stochastic functional constraints that are i.i.d. generated at each round and are disclosed to the decision maker only after the decision is made. This formulation arises naturally when decisions are restricted by stochastic environments or deterministic environments with noisy observations. It also includes many important problems as special case, such as OCO with long term constraints, stochastic constrained convex optimization, and deterministic constrained convex optimization. To solve this problem, this paper proposes a new algorithm that achieves $O(\sqrt{T})$ expected regret and constraint violations and $O(\sqrt{T}\log(T))$ high probability regret and constraint violations. Experiments on a real-world data center scheduling problem further verify the performance of the new algorithm.

## 1   Introduction

Online convex optimization (OCO) is a multi-round learning process with arbitrarily-varying convex loss functions where the decision maker has to choose decision $x(t) \in \mathcal{X}$ before observing the corresponding loss function $f^t(\cdot)$. For a fixed time horizon $T$, define the *regret* of a learning algorithm with respect to the best fixed decision in hindsight (with full knowledge of all loss functions) as

$$\text{regret}(T) = \sum_{t=1}^{T} f^t(\mathbf{x}(t)) - \min_{\mathbf{x} \in \mathcal{X}} \sum_{t=1}^{T} f^t(\mathbf{x}).$$

The goal of OCO is to develop dynamic learning algorithms such that regret grows sub-linearly with respect to $T$. The setting of OCO is introduced in a series of work [3, 14, 9, 29] and is formalized in [29]. OCO has gained considerable amount of research interest recently with various applications such as online regression, prediction with expert advice, online ranking, online shortest paths, and portfolio selection. See [23, 11] for more applications and background.

In [29], Zinkevich shows $O(\sqrt{T})$ regret can be achieved by using an online gradient descent (OGD) update given by

$$\mathbf{x}(t+1) = \mathcal{P}_{\mathcal{X}}\big[\mathbf{x}(t) - \gamma \nabla f^t(\mathbf{x}(t))\big] \tag{1}$$

where $\nabla f^t(\cdot)$ is a subgradient of $f^t(\cdot)$ and $\mathcal{P}_{\mathcal{X}}[\cdot]$ is the projection onto set $\mathcal{X}$. Hazan et al. in [12] show that better regret is possible under the assumption that each loss function is strongly convex but $O(\sqrt{T})$ is the best possible if no additional assumption is imposed.

It is obvious that Zinkevich's OGD in (1) requires the full knowledge of set $\mathcal{X}$ and low complexity of the projection $\mathcal{P}_{\mathcal{X}}[\cdot]$. However, in practice, the constraint set $\mathcal{X}$, which is often described by

---

[*]This work is supported in part by grant NSF CCF-1718477.

many functional inequality constraints, can be time varying and may not be fully disclosed to the decision maker. In [18], Mannor et al. extend OCO by considering time-varying constraint functions $g^t(\mathbf{x})$ which can arbitrarily vary and are only disclosed to us after each $\mathbf{x}(t)$ is chosen. In this setting, Mannor et al. in [18] explore the possibility of designing learning algorithms such that regret grows sub-linearly and $\limsup_{T\to\infty} \frac{1}{T}\sum_{t=1}^T g^t(\mathbf{x}(t)) \le 0$, i.e., the (cumulative) constraint violation $\sum_{t=1}^T g^t(\mathbf{x}(t))$ also grows sub-linearly. Unfortunately, Mannor et al. in [18] prove that this is impossible even when both $f^t(\cdot)$ and $g^t(\cdot)$ are simple linear functions.

Given the impossibility results shown by Mannor et al. in [18], this paper considers OCO where constraint functions $g^t(\mathbf{x})$ are not arbitrarily varying but independently and identically distributed (i.i.d.) generated from an unknown probability model (and functions $f^t(\mathbf{x})$ are still arbitrarily varying and possibly non-i.i.d.). Specifically, this paper considers *online convex optimization (OCO) with stochastic constraint* $\mathcal{X} = \{\mathbf{x} \in \mathcal{X}_0 : \mathbb{E}_\omega[g_k(\mathbf{x};\omega)] \le 0, k \in \{1,2,\ldots,m\}\}$ where $\mathcal{X}_0$ is a known fixed set; the expressions of stochastic constraints $\mathbb{E}_\omega[g_k(\mathbf{x};\omega)]$ (involving expectations with respect to $\omega$ from an unknown distribution) are unknown; and subscripts $k \in \{1,2,\ldots,m\}$ indicate the possibility of multiple functional constraints. In OCO with stochastic constraints, the decision maker receives loss function $f^t(\mathbf{x})$ and i.i.d. constraint function realizations $g_k^t(\mathbf{x}) \triangleq g_k(\mathbf{x};\omega(t))$ at each round $t$. However, the expressions of $g_k^t(\cdot)$ and $f^t(\cdot)$ are disclosed to the decision maker only after decision $\mathbf{x}(t) \in \mathcal{X}_0$ is chosen. This setting arises naturally when decisions are restricted by stochastic environments or deterministic environments with noisy observations. For example, if we consider online routing (with link capacity constraints) in wireless networks [18], each link capacity is not a fixed constant (as in wireline networks) but an i.i.d. random variable since wireless channels are stochastically time-varying by nature [25]. OCO with stochastic constraints also covers important special cases such as OCO with long term constraints [16, 5, 13], stochastic constrained convex optimization [17] and deterministic constrained convex optimization [21].

Let $\mathbf{x}^* = \operatorname{argmin}_{\{\mathbf{x} \in \mathcal{X}_0 : \mathbb{E}[g_k(\mathbf{x};\omega)] \le 0, \forall k \in \{1,2,\ldots,m\}\}} \sum_{t=1}^T f^t(\mathbf{x})$ be the best fixed decision in hindsight (knowing all loss functions $f^t(\mathbf{x})$ and the distribution of stochastic constraint functions $g_k(\mathbf{x};\omega)$). Thus, $\mathbf{x}^*$ minimizes the $T$-round cumulative loss and satisfies all stochastic constraints in expectation, which also implies $\limsup_{T\to\infty} \frac{1}{T}\sum_{t=1}^T g_k^t(\mathbf{x}^*) \le 0$ almost surely by the strong law of large numbers. Our goal is to develop dynamic learning algorithms that guarantee both regret $\sum_{t=1}^T f^t(\mathbf{x}(t)) - \sum_{t=1}^T f^t(\mathbf{x}^*)$ and constraint violations $\sum_{t=1}^T g_k^t(\mathbf{x}(t))$ grow sub-linearly.

Note that Zinkevich's algorithm in (1) is not applicable to OCO with stochastic constraints since $\mathcal{X}$ is unknown and it can happen that $\mathcal{X}(t) = \{\mathbf{x} \in \mathcal{X}_0 : g_k(\mathbf{x};\omega(t)) \le 0, \forall k \in \{1,2,\ldots,m\}\} = \emptyset$ for certain realizations $\omega(t)$, such that projections $\mathcal{P}_{\mathcal{X}}[\cdot]$ or $\mathcal{P}_{\mathcal{X}(t)}[\cdot]$ required in (1) are not even well-defined.

**Our Contributions:** This paper solves online convex optimization with stochastic constraints. In particular, we propose a new learning algorithm that is proven to achieve $O(\sqrt{T})$ expected regret and constraint violations and $O(\sqrt{T}\log(T))$ high probability regret and constraint violations. The proposed new algorithm also improves upon state-of-the-art results in the following special cases:

- *OCO with long term constraints:* This is a special case where each $g_k^t(\mathbf{x}) \equiv g_k(\mathbf{x})$ is known and does not depend on time. Note that $\mathcal{X} = \{\mathbf{x} \in \mathcal{X}_0 : g_k(\mathbf{x}) \le 0, \forall k \in \{1,2,\ldots,m\}\}$ can be complicated while $\mathcal{X}_0$ might be a simple hypercube. To avoid high complexity involved in the projection onto $\mathcal{X}$ as in Zinkevich's algorithm, work in [16, 5, 13] develops low complexity algorithms that use projections onto a simpler set $\mathcal{X}_0$ by allowing $g_k(\mathbf{x}(t)) > 0$ for certain rounds but ensuring $\limsup_{T\to\infty} \frac{1}{T}\sum_{t=1}^T g_k(\mathbf{x}(t)) \le 0$. The best existing performance is $O(T^{\max\{\beta,1-\beta\}})$ regret and $O(T^{1-\beta/2})$ constraint violations where $\beta \in (0,1)$ is an algorithm parameter [13]. This gives $O(\sqrt{T})$ regret with worse $O(T^{3/4})$ constraint violations or $O(\sqrt{T})$ constraint violations with worse $O(T)$ regret. In contrast, our algorithm, which only uses projections onto $\mathcal{X}_0$ as shown in Lemma 1, can achieve $O(\sqrt{T})$ regret and $O(\sqrt{T})$ constraint violations simultaneously. Note that by adapting the methodology presented in this paper, our other work [27] developed a different algorithm that can only solve the special case problem "OCO with long term constraints" but can achieve $O(\sqrt{T})$ regret and $O(1)$ constraint violations.

- *Stochastic constrained convex optimization:* This is a special case where each $f^t(\mathbf{x})$ is i.i.d. generated from an unknown distribution. This problem has many applications in operations research and machine learning such as Neyman-Pearson classification and risk-mean portfolio.

The work [17] develops a (batch) offline algorithm that produces a solution with high probability performance guarantees only after sampling the problems for sufficiently many times. That is, during the process of sampling, there is no performance guarantee. The work [15] proposes a stochastic approximation based (batch) offline algorithm for stochastic convex optimization with one single stochastic functional inequality constraint. In contrast, our algorithm is an online algorithm with online performance guarantees and can deal with an arbitrary number of stochastic constraints.

- *Deterministic constrained convex optimization:* This is a special case where each $f^t(\mathbf{x}) \equiv f(\mathbf{x})$ and $g_k^t(\mathbf{x}) \equiv g_k(\mathbf{x})$ are known and do not depend on time. In this case, the goal is to develop a fast algorithm that converges to a good solution (with a small error) with a few number of iterations; and our algorithm with $O(\sqrt{T})$ regret and constraint violations is equivalent to an iterative numerical algorithm with $O(1/\sqrt{T})$ convergence rate. Our algorithm is subgradient based and does not require the smoothness or differentiability of the convex program. The primal-dual subgradient method considered in [19] has the same $O(1/\sqrt{T})$ convergence rate but requires an upper bound of optimal Lagrange multipliers, which is usually unknown in practice.

## 2 Formulation and New Algorithm

Let $\mathcal{X}_0$ be a known fixed compact convex set. Let $f^t(\mathbf{x})$ be a sequence of arbitrarily-varying convex functions. Let $g_k(\mathbf{x}; \omega(t)), k \in \{1, 2, \ldots, m\}$ be sequences of functions that are i.i.d. realizations of stochastic constraint functions $\tilde{g}_k(\mathbf{x}) \triangleq \mathbb{E}_\omega[g_k(\mathbf{x}; \omega)]$ with random variable $\omega \in \Omega$ from an unknown distribution. That is, $\omega(t)$ are i.i.d. samples of $\omega$. Assume that each $f^t(\cdot)$ is independent of all $\omega(\tau)$ with $\tau \geq t + 1$ so that we are unable to predict future constraint functions based on the knowledge of the current loss function. For each $\omega \in \Omega$, we assume $g_k(\mathbf{x}; \omega)$ are convex with respect to $\mathbf{x} \in \mathcal{X}_0$. At the beginning of each round $t$, neither the loss function $f^t(\mathbf{x})$ nor the constraint function realizations $g_k(\mathbf{x}; \omega(t))$ are known to the decision maker. However, the decision maker still needs to make a decision $\mathbf{x}(t) \in \mathcal{X}_0$ for round $t$; and after that $f^t(\mathbf{x})$ and $g_k(\mathbf{x}, \omega(t))$ are disclosed to the decision maker at the end of round $t$.

For convenience, we often suppress the dependence of each $g_k(\mathbf{x}; \omega(t))$ on $\omega(t)$ and write $g_k^t(\mathbf{x}) = g_k(\mathbf{x}; \omega(t))$. Recall $\tilde{g}_k(\mathbf{x}) = \mathbb{E}_\omega[g_k(\mathbf{x}; \omega)]$ where the expectation is with respect to $\omega$. Define $\mathcal{X} = \{\mathbf{x} \in \mathcal{X}_0 : \tilde{g}_k(\mathbf{x}) = \mathbb{E}[g_k(\mathbf{x}; \omega)] \leq 0, \forall k \in \{1, 2, \ldots, m\}\}$. We further define the stacked vector of multiple functions $g_1^t(\mathbf{x}), \ldots, g_m^t(\mathbf{x})$ as $\mathbf{g}^t(\mathbf{x}) = [g_1^t(\mathbf{x}), \ldots, g_m^t(\mathbf{x})]^\mathsf{T}$ and define $\tilde{\mathbf{g}}(\mathbf{x}) = [\mathbb{E}_\omega[g_1(\mathbf{x}; \omega)], \ldots, \mathbb{E}_\omega[g_m(\mathbf{x}; \omega)]]^\mathsf{T}$. We use $\|\cdot\|$ to denote the Euclidean norm for a vector. Throughout this paper, we have the following assumptions:

**Assumption 1** (Basic Assumptions).

- *Loss functions $f^t(\mathbf{x})$ and constraint functions $g_k(\mathbf{x}; \omega)$ have bounded subgradients on $\mathcal{X}_0$. That is, there exists $D_1 > 0$ and $D_2 > 0$ such that $\|\nabla f^t(\mathbf{x})\| \leq D_1$ for all $\mathbf{x} \in \mathcal{X}_0$ and all $t \in \{0, 1, \ldots\}$ and $\|\nabla g_k(\mathbf{x}; \omega)\| \leq D_2$ for all $\mathbf{x} \in \mathcal{X}_0$, all $\omega \in \Omega$ and all $k \in \{1, 2, \ldots, m\}$.[2]*
- *There exists constant $G > 0$ such that $\|\mathbf{g}(\mathbf{x}; \omega)\| \leq G$ for all $\mathbf{x} \in \mathcal{X}_0$ and all $\omega \in \Omega$.*
- *There exists constant $R > 0$ such that $\|\mathbf{x} - \mathbf{y}\| \leq R$ for all $\mathbf{x}, \mathbf{y} \in \mathcal{X}_0$.*

**Assumption 2** (The Slater Condition). *There exists $\epsilon > 0$ and $\hat{\mathbf{x}} \in \mathcal{X}_0$ such that $\tilde{g}_k(\hat{\mathbf{x}}) = \mathbb{E}_\omega[g_k(\hat{\mathbf{x}}; \omega)] \leq -\epsilon$ for all $k \in \{1, 2, \ldots, m\}$.*

### 2.1 New Algorithm

Now consider the following algorithm described in Algorithm 1. This algorithm chooses $\mathbf{x}(t + 1)$ as the decision for round $t + 1$ based on $f^t(\cdot)$ and $\mathbf{g}^t(\cdot)$ without requiring $f^{t+1}(\cdot)$ or $\mathbf{g}^{t+1}(\cdot)$.

For each stochastic constraint function $g_k(\mathbf{x}; \omega)$, we introduce $Q_k(t)$ and call it a virtual queue since its dynamic is similar to a queue dynamic. The next lemma summarizes that $\mathbf{x}(t + 1)$ update in (2) can be implemented via a simple projection onto $\mathcal{X}_0$.

**Lemma 1.** *The $\mathbf{x}(t + 1)$ update in (2) is given by $\mathbf{x}(t + 1) = \mathcal{P}_{\mathcal{X}_0}\big[\mathbf{x}(t) - \frac{1}{2\alpha}\mathbf{d}(t)\big]$, where $\mathbf{d}(t) = V\nabla f^t(\mathbf{x}(t)) + \sum_{k=1}^{m} Q_k(t)\nabla g_k^t(\mathbf{x}(t))$ and $\mathcal{P}_{\mathcal{X}_0}[\cdot]$ is the projection onto convex set $\mathcal{X}_0$.*

**Algorithm 1**

Let $V > 0$ and $\alpha > 0$ be constant algorithm parameters. Choose $\mathbf{x}(1) \in \mathcal{X}_0$ arbitrarily and let $Q_k(1) = 0, \forall k \in \{1, 2, \ldots, m\}$. At the end of each round $t \in \{1, 2, \ldots\}$, observe $f^t(\cdot)$ and $\mathbf{g}^t(\cdot)$ and do the following:

- Choose $\mathbf{x}(t+1)$ that solves

$$\min_{\mathbf{x} \in \mathcal{X}_0} \Big\{ V[\nabla f^t(\mathbf{x}(t))]^\mathsf{T}[\mathbf{x} - \mathbf{x}(t)] + \sum_{k=1}^m Q_k(t)[\nabla g_k^t(\mathbf{x}(t))]^\mathsf{T}[\mathbf{x} - \mathbf{x}(t)] + \alpha \|\mathbf{x} - \mathbf{x}(t)\|^2 \Big\} \quad (2)$$

  as the decision for the next round $t + 1$, where $\nabla f^t(\mathbf{x}(t))$ is a subgradient of $f^t(\mathbf{x})$ at point $\mathbf{x} = \mathbf{x}(t)$ and $\nabla g_k^t(\mathbf{x}(t))$ is a subgradient of $g_k^t(\mathbf{x})$ at point $\mathbf{x} = \mathbf{x}(t)$.
- Update each virtual queue $Q_k(t+1), \forall k \in \{1, 2, \ldots, m\}$ via

$$Q_k(t+1) = \max \big\{ Q_k(t) + g_k^t(\mathbf{x}(t)) + [\nabla g_k^t(\mathbf{x}(t))]^\mathsf{T}[\mathbf{x}(t+1) - \mathbf{x}(t)], 0 \big\}, \quad (3)$$

  where $\max\{\cdot, \cdot\}$ takes the larger one between two elements.

---

*Proof.* The projection by definition is $\min_{\mathbf{x} \in \mathcal{X}_0} \|\mathbf{x} - [\mathbf{x}(t) - \frac{1}{2\alpha}\mathbf{d}(t)]\|^2$ and is equivalent to (2). $\quad\square$

## 2.2 Intuitions of Algorithm 1

Note that if there are no stochastic constraints $g_k^t(\mathbf{x})$, i.e., $\mathcal{X} = \mathcal{X}_0$, then Algorithm 1 has $Q_k(t) \equiv 0, \forall t$ and becomes Zinkevich's algorithm with $\gamma = \frac{V}{2\alpha}$ in (1) since

$$\mathbf{x}(t+1) \stackrel{(a)}{=} \operatorname*{argmin}_{\mathbf{x} \in \mathcal{X}_0} \Big\{ \underbrace{V[\nabla f^t(\mathbf{x}(t))]^\mathsf{T}[\mathbf{x} - \mathbf{x}(t)] + \alpha\|\mathbf{x} - \mathbf{x}(t)\|^2}_{\text{penalty}} \Big\} \stackrel{(b)}{=} \mathcal{P}_{\mathcal{X}_0}\Big[ \mathbf{x}(t) - \frac{V}{2\alpha}\nabla f^t(\mathbf{x}(t)) \Big] \quad (4)$$

where (a) follows from (2); and (b) follows from Lemma 1 by noting that $\mathbf{d}(t) = V\nabla f^t(\mathbf{x}(t))$. Call the term marked by an underbrace in (4) the *penalty*. Thus, Zinkevich's algorithm is to minimize the *penalty* term and is a special case of Algorithm 1 used to solve OCO over $\mathcal{X}_0$.

Let $\mathbf{Q}(t) = [Q_1(t), \ldots, Q_m(t)]^\mathsf{T}$ be the vector of virtual queue backlogs. Let $L(t) = \frac{1}{2}\|\mathbf{Q}(t)\|^2$ be a *Lyapunov function* and define *Lyapunov drift*

$$\Delta(t) = L(t+1) - L(t) = \frac{1}{2}[\|\mathbf{Q}(t+1)\|^2 - \|\mathbf{Q}(t)\|^2]. \quad (5)$$

The intuition behind Algorithm 1 is to choose $\mathbf{x}(t+1)$ to minimize an upper bound of the expression

$$\underbrace{\Delta(t)}_{\text{drift}} + \underbrace{V[\nabla f^t(\mathbf{x}(t))]^\mathsf{T}[\mathbf{x} - \mathbf{x}(t)] + \alpha\|\mathbf{x} - \mathbf{x}(t)\|^2}_{\text{penalty}} \quad (6)$$

The intention to minimize penalty is natural since Zinkevich's algorithm (for OCO without stochastic constraints) minimizes penalty, while the intention to minimize drift is motivated by observing that $g_k^t(\mathbf{x}(t))$ is accumulated into queue $Q_k(t+1)$ introduced in (3) such that we intend to have small queue backlogs. The drift $\Delta(t)$ can be complicated and is in general non-convex. The next lemma (proven in Supplement 7.1) provides a simple upper bound on $\Delta(t)$ and follows directly from (3).

**Lemma 2.** *At each round $t \in \{1, 2, \ldots\}$, Algorithm 1 guarantees*

$$\Delta(t) \le \sum_{k=1}^m Q_k(t) \big[ g_k^t(\mathbf{x}(t)) + [\nabla g_k^t(\mathbf{x}(t))]^\mathsf{T}[\mathbf{x}(t+1) - \mathbf{x}(t)] \big] + \frac{1}{2}[G + \sqrt{m}D_2 R]^2, \quad (7)$$

*where $m$ is the number of constraint functions; and $D_2, G$ and $R$ are defined in Assumption 1.*

At the end of round $t$, $\sum_{k=1}^m Q_k(t)g_k^t(\mathbf{x}(t)) + \frac{1}{2}[G + \sqrt{m}D_2 R]^2$ is a given constant that is not affected by decision $\mathbf{x}(t+1)$. The algorithm decision in (2) is now transparent: $\mathbf{x}(t+1)$ is chosen to minimize the drift-plus-penalty expression (6), where $\Delta(t)$ is approximated by the bound in (7).

## 2.3 Preliminary Analysis and More Intuitions of Algorithm 1

The next lemma (proven in Supplement 7.2) relates constraint violations and virtual queue values and follows directly from (3).

**Lemma 3.** *For any $T \geq 1$, Algorithm 1 guarantees $\sum_{t=1}^{T} g_k^t(\mathbf{x}(t)) \leq \|\mathbf{Q}(T+1)\| + D_2 \sum_{t=1}^{T} \|\mathbf{x}(t+1) - \mathbf{x}(t)\|, \forall k \in \{1, 2, \ldots, m\}$, where $D_2$ is defined in Assumption 1.*

Recall that function $h : \mathcal{X}_0 \to \mathbb{R}$ is said to be *c-strongly convex* if $h(x) - \frac{c}{2}\|x\|^2$ is convex over $\mathbf{x} \in \mathcal{X}_0$. It is easy to see that if $q : \mathcal{X}_0 \to \mathbb{R}$ is a convex function, then for any constant $c > 0$ and any vector $\mathbf{b}$, the function $q(x) + \frac{c}{2}\|\mathbf{x} - \mathbf{b}\|^2$ is *c-strongly convex*. Further, it is known that if $h : \mathcal{X} \to \mathbb{R}$ is a *c-strongly convex* function that is minimized at a point $\mathbf{x}^{min} \in \mathcal{X}_0$, then (see, for example, Corollary 1 in [28]):

$$h(\mathbf{x}^{min}) \leq h(\mathbf{x}) - \frac{c}{2}\|\mathbf{x} - \mathbf{x}^{min}\|^2 \quad \forall \mathbf{x} \in \mathcal{X}_0 \tag{8}$$

Note that the expression involved in minimization (2) in Algorithm 1 is strongly convex with modulus $2\alpha$ and $\mathbf{x}(t+1)$ is chosen to minimize it. Thus, the next lemma follows.

**Lemma 4.** *Let $\mathbf{z} \in \mathcal{X}_0$ be arbitrary. For all $t \geq 1$, Algorithm 1 guarantees*

$$V[\nabla f^t(\mathbf{x}(t))]^\mathsf{T}[\mathbf{x}(t+1) - \mathbf{x}(t)] + \sum_{k=1}^{m} Q_k(t)[\nabla g_k^t(\mathbf{x}(t))]^\mathsf{T}[\mathbf{x}(t+1) - \mathbf{x}(t)] + \alpha\|\mathbf{x}(t+1) - \mathbf{x}(t)\|^2$$

$$\leq V[\nabla f^t(\mathbf{x}(t))]^\mathsf{T}[\mathbf{z} - \mathbf{x}(t)] + \sum_{k=1}^{m} Q_k(t)[\nabla g_k^t(\mathbf{x}(t))]^\mathsf{T}[\mathbf{z} - \mathbf{x}(t)] + \alpha\|\mathbf{z} - \mathbf{x}(t)\|^2 - \alpha\|\mathbf{z} - \mathbf{x}(t+1)\|^2.$$

The next corollary follows by taking $\mathbf{z} = \mathbf{x}(t)$ in Lemma 4 and is proven in Supplement 7.3.

**Corollary 1.** *For all $t \geq 1$, Algorithm 1 guarantees $\|\mathbf{x}(t+1) - \mathbf{x}(t)\| \leq \frac{VD_1}{2\alpha} + \frac{\sqrt{m}D_2}{2\alpha}\|\mathbf{Q}(t)\|$.*

The next corollary follows directly from Lemma 3 and Corollary 1 and shows that constraint violations are ultimately bounded by sequence $\|\mathbf{Q}(t)\|, t \in \{1, 2, \ldots, T+1\}$.

**Corollary 2.** *For any $T \geq 1$, Algorithm 1 guarantees $\sum_{t=1}^{T} g_k^t(\mathbf{x}(t)) \leq \|\mathbf{Q}(T+1)\| + \frac{VTD_1D_2}{2\alpha} + \frac{\sqrt{m}D_2^2}{2\alpha} \sum_{t=1}^{T} \|\mathbf{Q}(t)\|, \forall k \in \{1, 2, \ldots, m\}$ where $D_1$ and $D_2$ are defined in Assumption 1.*

This corollary further justifies why Algorithm 1 intends to minimize drift $\Delta(t)$. As illustrated in the next section, controlled drift can often lead to boundedness of a stochastic process. Thus, the intuition of minimizing drift $\Delta(t)$ is to yield small $\|\mathbf{Q}(t)\|$ bounds.

# 3 Expected Performance Analysis of Algorithm 1

This section shows that if we choose $V = \sqrt{T}$ and $\alpha = T$ in Algorithm 1, then both expected regret and expected constraint violations are $O(\sqrt{T})$.

## 3.1 A Drift Lemma for Stochastic Processes

Let $\{Z(t), t \geq 0\}$ be a discrete time stochastic process adapted[3] to a filtration $\{\mathcal{F}(t), t \geq 0\}$. For example, $Z(t)$ can be a random walk, a Markov chain or a martingale. The drift analysis is the method of deducing properties, e.g., recurrence, ergodicity, or boundedness, about $Z(t)$ from its drift $\mathbb{E}[Z(t+1) - Z(t)|\mathcal{F}(t)]$. See [6, 10] for more discussions or applications on drift analysis. This paper proposes a new drift analysis lemma for stochastic processes as follows:

**Lemma 5.** *Let $\{Z(t), t \geq 0\}$ be a discrete time stochastic process adapted to a filtration $\{\mathcal{F}(t), t \geq 0\}$ with $Z(0) = 0$ and $\mathcal{F}(0) = \{\emptyset, \Omega\}$. Suppose there exists an integer $t_0 > 0$, real constants $\theta > 0$, $\delta_{\max} > 0$ and $0 < \zeta \leq \delta_{\max}$ such that*

$$|Z(t+1) - Z(t)| \leq \delta_{\max}, \tag{9}$$

$$\mathbb{E}[Z(t+t_0) - Z(t)|\mathcal{F}(t)] \leq \begin{cases} t_0\delta_{\max}, & \text{if } Z(t) < \theta \\ -t_0\zeta, & \text{if } Z(t) \geq \theta \end{cases}. \tag{10}$$

*hold for all $t \in \{1, 2, \ldots\}$. Then, the following holds*

1. *$\mathbb{E}[Z(t)] \leq \theta + t_0\delta_{\max} + t_0\frac{4\delta_{\max}^2}{\zeta}\log\left[\frac{8\delta_{\max}^2}{\zeta^2}\right], \forall t \in \{1, 2, \ldots\}$.*

2. *For any constant $0 < \mu < 1$, we have $Pr(Z(t) \geq z) \leq \mu, \forall t \in \{1, 2, \ldots\}$ where $z = \theta + t_0\delta_{\max} + t_0\frac{4\delta_{\max}^2}{\zeta}\log\left[\frac{8\delta_{\max}^2}{\zeta^2}\right] + t_0\frac{4\delta_{\max}^2}{\zeta}\log(\frac{1}{\mu})$.*

The above lemma is proven in Supplement 7.4 and provides both expected and high probability bounds for stochastic processes based on a drift condition. It will be used to establish upper bounds of virtual queues $\|\mathbf{Q}(t)\|$, which further leads to expected and high probability constraint performance bounds of our algorithm. For a given stochastic process $Z(t)$, it is possible to show the drift condition (10) holds for multiple $t_0$ with different $\zeta$ and $\theta$. In fact, we will show in Lemma 7 that $\|\mathbf{Q}(t)\|$ yielded by Algorithm 1 satisfies (10) for any integer $t_0 > 0$ by selecting $\zeta$ and $\theta$ according to $t_0$. One-step drift conditions, corresponding to the special case $t_0 = 1$ of Lemma 5, have been previously considered in [10, 20]. However, Lemma 5 (with general $t_0 > 0$) allows us to choose the best $t_0$ in performance analysis such that sublinear regret and constraint violation bounds are possible.

### 3.2  Expected Constraint Violation Analysis

Define filtration $\{\mathcal{W}(t), t \geq 0\}$ with $\mathcal{W}(0) = \{\emptyset, \Omega\}$ and $\mathcal{W}(t) = \sigma(\omega(1), \ldots, \omega(t))$ being the $\sigma$-algebra generated by random samples $\{\omega(1), \ldots, \omega(t)\}$ up to round $t$. From the update rule in Algorithm 1, we observe that $\mathbf{x}(t+1)$ is a deterministic function of $f^t(\cdot), \mathbf{g}(\cdot; \omega(t))$ and $\mathbf{Q}(t)$ where $\mathbf{Q}(t)$ is further a deterministic function of $\mathbf{Q}(t-1), \mathbf{g}(\cdot; \omega(t-1)), \mathbf{x}(t)$ and $\mathbf{x}(t-1)$. By inductions, it is easy to show that $\sigma(\mathbf{x}(t)) \subseteq \mathcal{W}(t-1)$ and $\sigma(\mathbf{Q}(t)) \subseteq \mathcal{W}(t-1)$ for all $t \geq 1$ where $\sigma(Y)$ denotes the $\sigma$-algebra generated by random variable $Y$. For fixed $t \geq 1$, since $\mathbf{Q}(t)$ is fully determined by $\omega(\tau), \tau \in \{1, 2, \ldots, t-1\}$ and $\omega(t)$ are i.i.d., we know $\mathbf{g}^t(\mathbf{x})$ is independent of $\mathbf{Q}(t)$. This is formally summarized in the next lemma.

**Lemma 6.** *If $\mathbf{x}^* \in \mathcal{X}_0$ satisfies $\tilde{\mathbf{g}}(\mathbf{x}^*) = \mathbb{E}_\omega[\mathbf{g}(\mathbf{x}^*; \omega)] \leq \mathbf{0}$, then Algorithm 1 guarantees:*

$$\mathbb{E}[Q_k(t) g_k^t(\mathbf{x}^*)] \leq 0, \forall k \in \{1, 2, \ldots, m\}, \forall t \geq 1. \tag{11}$$

*Proof.*  Fix $k \in \{1, 2, \ldots, m\}$ and $t \geq 1$. Since $g_k^t(\mathbf{x}^*) = g_k(\mathbf{x}^*; \omega(t))$ is independent of $Q_k(t)$, which is determined by $\{\omega(1), \ldots, \omega(t-1)\}$, it follows that $\mathbb{E}[Q_k(t) g_k^t(\mathbf{x}^*)] = \mathbb{E}[Q_k(t)]\mathbb{E}[g_k^t(\mathbf{x}^*)] \overset{(a)}{\leq} 0$, where (a) follows from the fact that $\mathbb{E}[g_k^t(\mathbf{x}^*)] \leq 0$ and $Q_k(t) \geq 0$.  □

To establish a bound on constraint violations, by Corollary 2, it suffices to derive upper bounds for $\|\mathbf{Q}(t)\|$. In this subsection, we derive upper bounds for $\|\mathbf{Q}(t)\|$ by applying the new drift lemma (Lemma 5) developed at the beginning of this section. The next lemma shows that random process $Z(t) = \|\mathbf{Q}(t)\|$ satisfies the conditions in Lemma 5.

**Lemma 7.** *Let $t_0 > 0$ be an arbitrary integer. At each round $t \in \{1, 2, \ldots,\}$ in Algorithm 1, the following holds*

$$\left| \|\mathbf{Q}(t+1)\| - \|\mathbf{Q}(t)\| \right| \leq G + \sqrt{m}D_2 R, \quad and$$

$$\mathbb{E}[\|\mathbf{Q}(t+t_0)\| - \|\mathbf{Q}(t)\| \big| \mathcal{W}(t-1)] \leq \begin{cases} t_0(G + \sqrt{m}D_2 R), & if \ \|\mathbf{Q}(t)\| < \theta \\ -t_0 \frac{\epsilon}{2}, & if \ \|\mathbf{Q}(t)\| \geq \theta \end{cases},$$

*where $\theta = \frac{\epsilon}{2} t_0 + (G + \sqrt{m}D_2 R) t_0 + \frac{2\alpha R^2}{t_0 \epsilon} + \frac{2V D_1 R + [G + \sqrt{m}D_2 R]^2}{\epsilon}$, $m$ is the number of constraint functions; $D_1, D_2, G$ and $R$ are defined in Assumption 1; and $\epsilon$ is defined in Assumption 2. (Note that $\epsilon < G$ by the definition of $G$.)*

Lemma 7 (proven in Supplement 7.5) allows us to apply Lemma 5 to random process $Z(t) = \|\mathbf{Q}(t)\|$ and obtain $\mathbb{E}[\|\mathbf{Q}(t)\|] = O(\sqrt{T}), \forall t$ by taking $t_0 = \lceil \sqrt{T} \rceil$, $V = \sqrt{T}$ and $\alpha = T$, where $\lceil \sqrt{T} \rceil$ represents the smallest integer no less than $\sqrt{T}$. By Corollary 2, this further implies the expected constraint violation bound $\mathbb{E}[\sum_{t=1}^T g_k(\mathbf{x}(t))] \leq O(\sqrt{T})$ as summarized in the next theorem.

**Theorem 1** (Expected Constraint Violation Bound). *If $V = \sqrt{T}$ and $\alpha = T$ in Algorithm 1, then for all $T \geq 1$, we have*

$$\mathbb{E}[\sum_{t=1}^T g_k^t(\mathbf{x}(t))] \leq O(\sqrt{T}), \forall k \in \{1, 2, \ldots, m\}. \tag{12}$$

*where the expectation is taken with respect to all $\omega(t)$.*

*Proof.* Define random process $Z(t)$ with $Z(0) = 0$ and $Z(t) = \|\mathbf{Q}(t)\|, t \geq 1$ and filtration $\mathcal{F}(t)$ with $\mathcal{F}(0) = \{\emptyset, \Omega\}$ and $\mathcal{F}(t) = \mathcal{W}(t-1), t \geq 1$. Note that $Z(t)$ is adapted to $\mathcal{F}(t)$. By

Lemma 7, $Z(t)$ satisfies the conditions in Lemma 5 with $\delta_{\max} = G + \sqrt{m}D_2 R$, $\zeta = \frac{\epsilon}{2}$ and $\theta = \frac{\epsilon}{2}t_0 + (G + \sqrt{m}D_2 R)t_0 + \frac{2\alpha R^2}{t_0 \epsilon} + \frac{2VD_1 R + [G + \sqrt{m}D_2 R]^2}{\epsilon}$. Thus, by part (1) of Lemma 5, for all $t \in \{1, 2, \dots\}$, we have $\mathbb{E}[\|\mathbf{Q}(t)\|] \leq \frac{\epsilon}{2}t_0 + 2(G + \sqrt{m}D_2 R)t_0 + \frac{2\alpha R^2}{t_0 \epsilon} + \frac{2VD_1 R + [G + \sqrt{m}D_2 R]^2}{\epsilon} + t_0 \frac{8[G + \sqrt{m}D_2 R]^2}{\epsilon} \log[\frac{32[G + \sqrt{m}D_2 R]^2}{\epsilon^2}]$. Taking $t_0 = \lceil \sqrt{T} \rceil$, $V = \sqrt{T}$ and $\alpha = T$, we have $\mathbb{E}[\|\mathbf{Q}(t)\|] \leq O(\sqrt{T})$ for all $t \in \{1, 2, \dots\}$.

Fix $T \geq 1$. By Corollary 2 (with $V = \sqrt{T}$ and $\alpha = T$), we have $\sum_{t=1}^{T} g_k^t(\mathbf{x}(t)) \leq \|\mathbf{Q}(T + 1)\| + \frac{\sqrt{T}D_1 D_2}{2} + \frac{\sqrt{m}D_2^2}{2T} \sum_{t=1}^{T} \|\mathbf{Q}(t)\|, \forall k \in \{1, 2, \dots, m\}$. Taking expectations on both sides and substituting $\mathbb{E}[\|\mathbf{Q}(t)\|] = O(\sqrt{T}), \forall t$ into it yields $\mathbb{E}[\sum_{t=1}^{T} g_k^t(\mathbf{x}(t))] \leq O(\sqrt{T})$. □

### 3.3 Expected Regret Analysis

The next lemma (proven in Supplement 7.6) refines Lemma 4 and is useful to analyze the regret.

**Lemma 8.** *Let $\mathbf{z} \in \mathcal{X}_0$ be arbitrary. For all $T \geq 1$, Algorithm 1 guarantees*

$$\sum_{t=1}^{T} f^t(\mathbf{x}(t)) \leq \sum_{t=1}^{T} f^t(\mathbf{z}) + \underbrace{\frac{\alpha}{V}R^2 + \frac{VD_1^2}{4\alpha}T + \frac{1}{2}[G + \sqrt{m}D_2 R]^2 \frac{T}{V}}_{(I)} + \underbrace{\frac{1}{V}\sum_{t=1}^{T}[\sum_{k=1}^{m} Q_k(t)g_k^t(\mathbf{z})]}_{(II)} \quad (13)$$

*where $m$ is the number of constraint functions; and $D_1, D_2, G$ and $R$ are defined in Assumption 1.*

Note that if we take $V = \sqrt{T}$ and $\alpha = T$, then term (I) in (13) is $O(\sqrt{T})$. Recall that the expectation of term (II) in (13) with $\mathbf{z} = \mathbf{x}^*$ is non-positive by Lemma 6. The expected regret bound of Algorithm 1 follows by taking expectations on both sides of (13) and is summarized in the next theorem.

**Theorem 2** (Expected Regret Bound). *Let $\mathbf{x}^* \in \mathcal{X}_0$ be any fixed solution that satisfies $\tilde{\mathbf{g}}(\mathbf{x}^*) \leq \mathbf{0}$, e.g., $\mathbf{x}^* = \text{argmin}_{\mathbf{x} \in \mathcal{X}} \sum_{t=1}^{T} f^t(\mathbf{x})$. If $V = \sqrt{T}$ and $\alpha = T$ in Algorithm 1, then for all $T \geq 1$,*

$$\mathbb{E}[\sum_{t=1}^{T} f^t(\mathbf{x}(t))] \leq \mathbb{E}[\sum_{t=1}^{T} f^t(\mathbf{x}^*)] + O(\sqrt{T}).$$

*where the expectation is taken with respect to all $\omega(t)$.*

*Proof.* Fix $T \geq 1$. Taking $\mathbf{z} = \mathbf{x}^*$ in Lemma 8 yields $\sum_{t=1}^{T} f^t(\mathbf{x}(t)) \leq \sum_{t=1}^{T} f^t(\mathbf{x}^*) + \frac{\alpha}{V}R^2 + \frac{VD_1^2}{4\alpha}T + \frac{1}{2}[G + \sqrt{m}D_2 R]^2 \frac{T}{V} + \frac{1}{V}\sum_{t=1}^{T}[\sum_{k=1}^{m} Q_k(t)g_k^t(\mathbf{x}^*)]$. Taking expectations on both sides and using (11) yields $\sum_{t=1}^{T} \mathbb{E}[f^t(\mathbf{x}(t))] \leq \sum_{t=1}^{T} \mathbb{E}[f^t(\mathbf{x}^*)] + R^2 \frac{\alpha}{V} + \frac{D_1^2}{4}\frac{V}{\alpha}T + \frac{1}{2}[G + \sqrt{m}D_2 R]^2 \frac{T}{V}$. Taking $V = \sqrt{T}$ and $\alpha = T$ yields $\sum_{t=1}^{T} \mathbb{E}[f^t(\mathbf{x}(t))] \leq \sum_{t=1}^{T} \mathbb{E}[f^t(\mathbf{x}^*)] + O(\sqrt{T})$. □

### 3.4 Special Case Performance Guarantees

Theorems 1 and 2 provide expected performance guarantees of Algorithm 1 for OCO with stochastic constraints. The results further imply the performance guarantees in the following special cases:

- **OCO with long term constraints**: In this case, $g_k(\mathbf{x}; \omega(t)) \equiv g_k(\mathbf{x})$ and there is no randomness. Thus, the expectations in Theorems 1 and 2 disappear. For this problem, Algorithm 1 can achieve $O(\sqrt{T})$ (deterministic) regret and $O(\sqrt{T})$ (deterministic) constraint violations.

- **Stochastic constrained convex optimization**: Note that i.i.d. time-varying $f(\mathbf{x}; \omega(t))$ is a special case of arbitrarily-varying $f^t(\mathbf{x})$ as considered in our OCO setting. Thus, Theorems 1 and 2 still hold when Algorithm 1 is applied to solve stochastic constrained convex optimization $\min_{\mathbf{x}}\{\mathbb{E}[f(\mathbf{x}; \omega)] : \mathbb{E}[g_k(\mathbf{x}; \omega)] \leq 0, \forall k \in \{1, 2, \dots, m\}, \mathbf{x} \in \mathcal{X}_0\}$ in an online fashion with i.i.d. realizations $\omega(t) \sim \omega$. Since Algorithm 1 chooses each $\mathbf{x}(t)$ without knowing $\omega(t)$, it follows that $\mathbf{x}(t)$ is independent of $\omega(t')$ for any $t' \geq t$ by the i.i.d. property of each $\omega(t)$. Fix $T > 0$, if we run Algorithm 1 for $T$ slots and use $\overline{\mathbf{x}}(T) = \frac{1}{T}\sum_{t=1}^{T} \mathbf{x}(t)$ as a fixed solution for any future slot $t' \geq T + 1$, then $\mathbb{E}[f(\overline{\mathbf{x}}(T); \omega(t'))] \overset{(a)}{\leq} \frac{1}{T}\sum_{t=1}^{T} \mathbb{E}[f(\mathbf{x}(t); \omega(t'))] \overset{(b)}{=} \frac{1}{T}\sum_{t=1}^{T} \mathbb{E}[f(\mathbf{x}(t); \omega(t))] \overset{(c)}{\leq} \frac{1}{T}\sum_{t=1}^{T} \mathbb{E}[f(\mathbf{x}^*; \omega(t))] + O(\frac{1}{\sqrt{T}}) \overset{(d)}{=} \mathbb{E}[f(\mathbf{x}^*; \omega(t'))] + O(\frac{1}{\sqrt{T}})$ and $\mathbb{E}[g_k(\overline{\mathbf{x}}(T); \omega(t'))] \overset{(a)}{\leq} \frac{1}{T}\sum_{t=1}^{T} \mathbb{E}[g_k(\overline{\mathbf{x}}(T); \omega(t'))] \overset{(b)}{=} \frac{1}{T}\sum_{t=1}^{T} \mathbb{E}[g_k(\mathbf{x}(t); \omega(t))] \overset{(c)}{\leq}$

$O(\frac{1}{\sqrt{T}}), \forall k \in \{1, 2, \ldots, m\}$ where (a) follows from Jensen's inequality and the fact that $\overline{\mathbf{x}}(T)$ is independent of $\omega(t')$; (b) follows because each $\mathbf{x}(t)$ is independent of both $\omega(t)$ and $\omega(t')$ and $\omega(t), \omega(t')$ are i.i.d. realizations of $\omega$; (c) follows from Theorems 1 and 2 by dividing both sides by $T$ and (d) follows because $\mathbb{E}[f(\mathbf{x}^*; \omega(t))] = \mathbb{E}[f(\mathbf{x}^*; \omega(t'))]$ for all $t \in \{1, \ldots, T\}$ by the i.i.d. property of each $\omega(t)$. Thus, if we use Algorithm 1 as a (batch) offline algorithm to solve stochastic constrained convex optimization, it has $O(1/\sqrt{T})$ convergence and ties with the algorithm developed in [15], which is by design a (batch) offline algorithm and can only solve stochastic optimization with a single constraint function.

- **Deterministic constrained convex optimization**: Similarly to OCO with long term constraints, the expectations in Theorems 1 and 2 disappear in this case since $f^t(\mathbf{x}) \equiv f(\mathbf{x})$ and $g_k(\mathbf{x}; \omega(t)) \equiv g_k(\mathbf{x})$. If we use $\overline{\mathbf{x}}(T) = \frac{1}{T} \sum_{t=1}^{T} \mathbf{x}(t)$ as the solution, then $f(\overline{\mathbf{x}}(T)) \leq f(\mathbf{x}^*) + O(\frac{1}{\sqrt{T}})$ and $g_k(\overline{\mathbf{x}}(T)) \leq O(\frac{1}{\sqrt{T}})$, which follows by dividing inequalities in Theorems 1 and 2 by $T$ on both sides and applying Jensen's inequality. Thus, Algorithm 1 solves deterministic constrained convex optimization with $O(\frac{1}{\sqrt{T}})$ convergence.

## 4 High Probability Performance Analysis

This section shows that if we choose $V = \sqrt{T}$ and $\alpha = T$ in Algorithm 1, then for any $0 < \lambda < 1$, with probability at least $1 - \lambda$, regret is $O(\sqrt{T} \log(T) \log^{1.5}(\frac{1}{\lambda}))$ and constraint violations are $O(\sqrt{T} \log(T) \log(\frac{1}{\lambda}))$.

### 4.1 High Probability Constraint Violation Analysis

Similarly to the expected constraint violation analysis, we can use part (2) of the new drift lemma (Lemma 5) to obtain a high probability bound of $\|\mathbf{Q}(t)\|$, which together with Corollary 2 leads to a high probability constraint violation bound summarized in Theorem 3 (proven in Supplement 7.7).

**Theorem 3** (High Probability Constraint Violation Bound). *Let $0 < \lambda < 1$ be arbitrary. If $V = \sqrt{T}$ and $\alpha = T$ in Algorithm 1, then for all $T \geq 1$ and all $k \in \{1, 2, \ldots, m\}$, we have*

$$Pr\Big( \sum_{t=1}^{T} g_k(\mathbf{x}(t)) \leq O\big(\sqrt{T} \log(T) \log(\frac{1}{\lambda})\big) \Big) \geq 1 - \lambda.$$

### 4.2 High Probability Regret Analysis

To obtain a high probability regret bound from Lemma 8, it remains to derive a high probability bound of term (II) in (13) with $\mathbf{z} = \mathbf{x}^*$. The main challenge is that term (II) is a supermartingale with unbounded differences (due to the possibly unbounded virtual queues $Q_k(t)$). Most concentration inequalities, e.g., the Hoeffding-Azuma inequality, used in high probability performance analysis of online algorithms are restricted to martingales/supermartingales with bounded differences. See for example [4, 2, 16]. The following lemma considers supermartingales with unbounded differences. Its proof (provided in Supplement 7.8) uses the truncation method to construct an auxiliary well-behaved supermartingale. Similar proof techniques are previously used in [26, 24] to prove different concentration inequalities for supermartingales/martingales with unbounded differences.

**Lemma 9.** *Let $\{Z(t), t \geq 0\}$ be a supermartingale adapted to a filtration $\{\mathcal{F}(t), t \geq 0\}$ with $Z(0) = 0$ and $\mathcal{F}(0) = \{\emptyset, \Omega\}$, i.e., $\mathbb{E}[Z(t+1)|\mathcal{F}(t)] \leq Z(t), \forall t \geq 0$. Suppose there exits a constant $c > 0$ such that $\{|Z(t+1) - Z(t)| > c\} \subseteq \{Y(t) > 0\}, \forall t \geq 0$, where $Y(t)$ is process with $Y(t)$ adapted to $\mathcal{F}(t)$ for all $t \geq 0$. Then, for all $z > 0$, we have*

$$Pr(Z(t) \geq z) \leq e^{-z^2/(2tc^2)} + \sum_{\tau=0}^{t-1} Pr(Y(\tau) > 0), \forall t \geq 1.$$

Note that if $\Pr(Y(t) > 0) = 0, \forall t \geq 0$, then $\Pr(\{|Z(t+1) - Z(t)| > c\}) = 0, \forall t \geq 0$ and $Z(t)$ is a supermartingale with differences bounded by $c$. In this case, Lemma 9 reduces to the conventional Hoeffding-Azuma inequality.

The next theorem (proven in Supplement 7.9) summarizes the high probability regret performance of Algorithm 1 and follows from Lemmas 5-9.

**Theorem 4** (High Probability Regret Bound). *Let $\mathbf{x}^* \in \mathcal{X}_0$ be any fixed solution that satisfies $\tilde{\mathbf{g}}(\mathbf{x}^*) \leq \mathbf{0}$, e.g., $\mathbf{x}^* = \operatorname{argmin}_{\mathbf{x} \in \mathcal{X}} \sum_{t=1}^{T} f^t(\mathbf{x})$. Let $0 < \lambda < 1$ be arbitrary. If $V = \sqrt{T}$ and $\alpha = T$ in Algorithm 1, then for all $T \geq 1$, we have*

$$Pr\Big( \sum_{t=1}^{T} f^t(\mathbf{x}(t)) \leq \sum_{t=1}^{T} f^t(\mathbf{x}^*) + O(\sqrt{T}\log(T)\log^{1.5}(\frac{1}{\lambda})) \Big) \geq 1 - \lambda.$$

## 5 Experiment: Online Job Scheduling in Distributed Data Centers

Consider a geo-distributed data center infrastructure consisting of one front-end job router and 100 geographically distributed servers, which are located at 10 different zones to form 10 clusters (10 servers in each cluster). See Fig. 1(a) for an illustration. The front-end job router receives job tasks and schedules them to different servers to fulfill the service. To serve the assigned jobs, each server purchases power (within its capacity) from its zone market. Electricity market prices can vary significantly across time and zones. For example, see Fig. 1(b) for a 5-minute average electricity price trace (between $05/01/2017$ and $05/10/2017$) at New York zone CENTRL [1]. This problem is to schedule jobs and control power levels at each server in real time such that all incoming jobs are served and electricity cost is minimized. In our experiment, each server power is adjusted every 5 minutes, which is called a slot. (In practice, server power can not be adjusted too frequently due to hardware restrictions and configuration delay.) Let $\mathbf{x}(t) = [x_1(t), \ldots, x_{100}(t)]$ be the power vector at slot $t$, where each $x_i(t)$ must be chosen from an interval $[x_i^{\min}, x_i^{\max}]$ restricted by the hardware, and the service rate at each server $i$ satisfies $\mu_i(t) = h_i(x_i(t))$, where $h_i(\cdot)$ is an increasing concave function. At each slot $t$, the job router schedules $\mu_i(t)$ amount of jobs to server $i$. The electricity cost at slot $t$ is $f^t(\mathbf{x}) = \sum_{i=1}^{100} c_i(t) x_i(t)$ where $c_i(t)$ is the electricity price at server $i$'s zone. We use $c_i(t)$ from real-world 5-minute average electricity price data at 10 different zones in New York city between $05/01/2017$ and $05/10/2017$ obtained from NYISO [1]. At each slot $t$, the incoming job is given by $\omega(t)$ and satisfies a Poisson distribution. Note that the amount of incoming jobs and electricity price $c_i(t)$ are unknown to us at the beginning of each slot $t$ but can be observed at the end of each slot. This is an example of OCO with stochastic constraints, where we aim to minimize the electricity cost subject to the constraint that incoming jobs must be served in time. In particular, at each round $t$, we receive loss function $f^t(\mathbf{x}(t))$ and constraint function $g^t(\mathbf{x}(t)) = \omega(t) - \sum_{i=1}^{100} h_i(x_i(t))$.

We compare our proposed algorithm with 3 baselines: (1) best fixed decision in hindsight; (2) react [8] and (3) low-power [22]. Both "react" and "low-power" are popular power control strategies used in distributed data centers. See Supplement 7.10 for more details of these 2 baselines and our experiment. Fig. 1(c)(d) plot the performance of 4 algorithms, where the running average is the time average up to the current slot. Fig. 1(c) compares electricity cost while Fig. 1(d) compares unserved jobs. (Unserved jobs accumulate if the service rate provided by an algorithm is less than the job arrival rate, i.e., the stochastic constraint is violated.) Fig. 1(c)(d) show that our proposed algorithm performs closely to the best fixed decision in hindsight over time, both in electricity cost and constraint violations. 'React" performs well in serving job arrivals but yields larger electricity cost, while "low-power" has low electricity cost but fails to serve job arrivals.

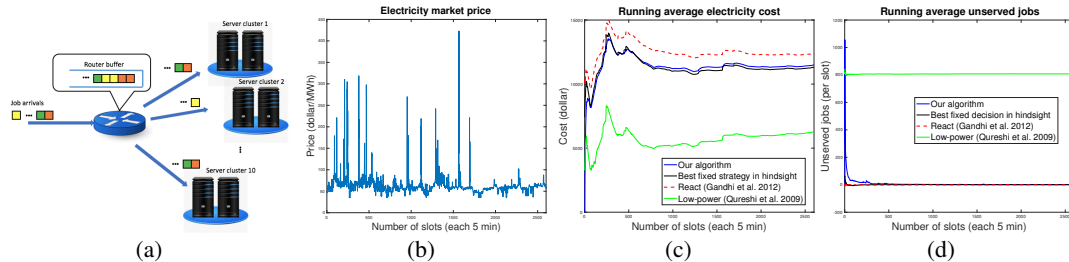

Figure 1: (a) Geo-distributed data center infrastructure; (b) Electricity market prices at zone CEN-TRAL New York; (c) Running average electricity cost; (d) Running average unserved jobs.

## 6 Conclusion

This paper studies OCO with stochastic constraints, where the objective function varies arbitrarily but the constraint functions are i.i.d. over time. A novel learning algorithm is developed that guarantees $O(\sqrt{T})$ expected regret and constraint violations and $O(\sqrt{T}\log(T))$ high probability regret and constraint violations.

## Footnotes

[2] The notation $\nabla h(\mathbf{x})$ is used to denote a subgradient of a convex function $h$ at the point $\mathbf{x}$.; it is the same as the gradient whenever the gradient exists.

[3]Random variable $Y$ is said to be adapted to $\sigma$-algebra $\mathcal{F}$ if $Y$ is $\mathcal{F}$-measurable. In this case, we often write $Y \in \mathcal{F}$. Similarly, random process $\{Z(t)\}$ is adapted to filtration $\{\mathcal{F}(t)\}$ if $Z(t) \in \mathcal{F}(t), \forall t$. See e.g. [7].

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
