[Supplementary Material]

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

# 7 Supplement

## 7.1 Proof of Lemma 2

Recall that for any $b \in \mathbb{R}$, if $a = \max\{b, 0\}$ then $a^2 \leq b^2$. Fix $k \in \{1, 2, \ldots, m\}$. The virtual queue update equation $Q_k(t+1) = \max\{Q_k(t) + g_k^t(\mathbf{x}(t)) + [\nabla g_k^t(\mathbf{x}(t))]^\mathsf{T}[\mathbf{x}(t+1) - \mathbf{x}(t)], 0\}$ implies that

$$
\begin{aligned}
\frac{1}{2}[Q_k(t+1)]^2 &\leq \frac{1}{2}\big[Q_k(t) + g_k^t(\mathbf{x}(t)) + [\nabla g_k^t(\mathbf{x}(t))]^\mathsf{T}[\mathbf{x}(t+1) - \mathbf{x}(t)]\big]^2 \\
&= \frac{1}{2}[Q_k(t)]^2 + Q_k(t)\big[g_k^t(\mathbf{x}(t)) + [\nabla g_k^t(\mathbf{x}(t))]^\mathsf{T}[\mathbf{x}(t+1) - \mathbf{x}(t)]\big] \\
&\quad + \frac{1}{2}\big[g_k^t(\mathbf{x}(t)) + [\nabla g_k^t(\mathbf{x}(t))]^\mathsf{T}[\mathbf{x}(t+1) - \mathbf{x}(t)]\big]^2 \\
&\overset{(a)}{=} \frac{1}{2}[Q_k(t)]^2 + Q_k(t)\big[g_k^t(\mathbf{x}(t)) + [\nabla g_k^t(\mathbf{x}(t))]^\mathsf{T}[\mathbf{x}(t+1) - \mathbf{x}(t)]\big] + \frac{1}{2}[h_k]^2, \quad (14)
\end{aligned}
$$

where (a) follows by defining $h_k = g_k^t(\mathbf{x}(t)) + [\nabla g_k^t(\mathbf{x}(t))]^\mathsf{T}[\mathbf{x}(t+1) - \mathbf{x}(t)]$.

Define $\mathbf{s} = [s_1, \ldots, s_m]^\mathsf{T}$, where $s_k = [\nabla g_k^t(\mathbf{x}(t))]^\mathsf{T}[\mathbf{x}(t+1) - \mathbf{x}(t)], \forall k \in \{1, 2, \ldots, m\}$; and $\mathbf{h} = [h_1, \ldots, h_m]^\mathsf{T} = \mathbf{g}^t(\mathbf{x}(t)) + \mathbf{s}$. Then,

$$
\|\mathbf{h}\| \overset{(a)}{\leq} \|\mathbf{g}^t(\mathbf{x}(t))\| + \|\mathbf{s}\| \overset{(b)}{\leq} G + \sqrt{\sum_{k=1}^{m} D_2^2 R^2} = G + \sqrt{m} D_2 R, \quad (15)
$$

where (a) follows from the triangle inequality; and (b) follows from the definition of Euclidean norm, the Cauchy-Schwarz inequality and Assumption 1.

Summing (14) over $k \in \{1, 2, \ldots, m\}$ yields

$$
\begin{aligned}
&\frac{1}{2}\|\mathbf{Q}(t+1)\|^2 \\
&\leq \frac{1}{2}\|\mathbf{Q}(t)\|^2 + \sum_{k=1}^{m} Q_k(t)\big[g_k^t(\mathbf{x}(t)) + [\nabla g_k^t(\mathbf{x}(t))]^\mathsf{T}[\mathbf{x}(t+1) - \mathbf{x}(t)]\big] + \frac{1}{2}\|\mathbf{h}\|^2 \\
&\overset{(a)}{\leq} \frac{1}{2}\|\mathbf{Q}(t)\|^2 + \sum_{k=1}^{m} Q_k(t)\big[g_k^t(\mathbf{x}(t)) + [\nabla g_k^t(\mathbf{x}(t))]^\mathsf{T}[\mathbf{x}(t+1) - \mathbf{x}(t)]\big] + \frac{1}{2}[G + \sqrt{m} D_2 R]^2,
\end{aligned}
$$

where (a) follows from (15). Rearranging the terms yields the desired result.

## 7.2 Proof of Lemma 3

Fix $k \in \{1, 2, \ldots, m\}$ and $T \geq 1$. For any $t \in \{0, 1, \ldots\}$, (3) in Algorithm 1 gives:

$$
\begin{aligned}
Q_k(t+1) &= \max\{Q_k(t) + g_k^t(\mathbf{x}(t)) + [\nabla g_k^t(\mathbf{x}(t))]^\mathsf{T}[\mathbf{x}(t+1) - \mathbf{x}(t)], 0\} \\
&\geq Q_k(t) + g_k^t(\mathbf{x}(t)) + [\nabla g_k^t(\mathbf{x}(t))]^\mathsf{T}[\mathbf{x}(t+1) - \mathbf{x}(t)] \\
&\overset{(a)}{\geq} Q_k(t) + g_k^t(\mathbf{x}(t)) - \|\nabla g_k^t(\mathbf{x}(t))\|\|\mathbf{x}(t+1) - \mathbf{x}(t)\| \\
&\overset{(b)}{\geq} Q_k(t) + g_k^t(\mathbf{x}(t)) - D_2\|\mathbf{x}(t+1) - \mathbf{x}(t)\|,
\end{aligned}
$$

where (a) follows from the Cauchy-Schwarz inequality and (b) follows from Assumption 1. Rearranging terms yields

$$
g_k^t(\mathbf{x}(t)) \leq Q_k(t+1) - Q_k(t) + D_2\|\mathbf{x}(t+1) - \mathbf{x}(t)\|.
$$

Summing over $t \in \{1, \ldots, T\}$ yields

$$\sum_{t=1}^{T} g_k^t(\mathbf{x}(t)) \leq Q_k(T+1) - Q_k(1) + D_2 \sum_{t=1}^{T} \|\mathbf{x}(t+1) - \mathbf{x}(t)\|$$

$$\stackrel{(a)}{=} Q_k(T+1) + D_2 \sum_{t=1}^{T} \|\mathbf{x}(t+1) - \mathbf{x}(t)\|$$

$$\leq \|\mathbf{Q}(T+1)\| + D_2 \sum_{t=1}^{T} \|\mathbf{x}(t+1) - \mathbf{x}(t)\|.$$

where (a) follows from the fact $Q_k(1) = 0$.

## 7.3 Proof of Corollary 1

Fix $t \geq 1$. Note that $\mathbf{x}(t) \in \mathcal{X}_0$. Taking $\mathbf{z} = \mathbf{x}(t)$ in Lemma 4 yields

$$V[\nabla f^t(\mathbf{x}(t))]^{\mathsf{T}}[\mathbf{x}(t+1) - \mathbf{x}(t)] + \sum_{k=1}^{m} Q_k(t)[\nabla g_k^t(\mathbf{x}(t))]^{\mathsf{T}}[\mathbf{x}(t+1) - \mathbf{x}(t)] + \alpha\|\mathbf{x}(t+1) - \mathbf{x}(t)\|^2$$

$$\leq -\alpha\|\mathbf{x}(t) - \mathbf{x}(t+1)\|^2.$$

Rearranging terms and cancelling common terms yields

$$2\alpha\|\mathbf{x}(t+1) - \mathbf{x}(t)\|^2$$

$$\leq -V[\nabla f^t(\mathbf{x}(t))]^{\mathsf{T}}[\mathbf{x}(t+1) - \mathbf{x}(t)] - \sum_{k=1}^{m} Q_k(t)\left[[\nabla g_k^t(\mathbf{x}(t))]^{\mathsf{T}}[\mathbf{x}(t+1) - \mathbf{x}(t)]\right]$$

$$\stackrel{(a)}{\leq} V\|\nabla f^t(\mathbf{x}(t))\|\|\mathbf{x}(t+1) - \mathbf{x}(t)\| + \|\mathbf{Q}(t)\|\sqrt{\sum_{k=1}^{m} \|\nabla g_k^t(\mathbf{x}(t))\|^2\|\mathbf{x}(t+1) - \mathbf{x}(t)\|^2}$$

$$\stackrel{(b)}{\leq} V D_1\|\mathbf{x}(t+1) - \mathbf{x}(t)\| + \sqrt{m} D_2\|\mathbf{Q}(t)\|\|\mathbf{x}(t+1) - \mathbf{x}(t)\|$$

where (a) follows by the Cauchy-Schwarz inequality (note that the second term on the right side applies the Cauchy-Schwarz inequality twice); and (b) follows from Assumption 1.

Thus, we have

$$\|\mathbf{x}(t+1) - \mathbf{x}(t)\| \leq \frac{V D_1}{2\alpha} + \frac{\sqrt{m} D_2}{2\alpha}\|\mathbf{Q}(t)\|.$$

## 7.4 Proof of Lemma 5

In this proof, we first establish an upper bound of $\mathbb{E}[e^{rZ(t)}]$ for some constant $r > 0$. Part (1) of this lemma follows by applying Jensen's inequality since $e^{rx}$ is convex with respect to $x$ when $r > 0$. Part (2) of this lemma follows directly from Markov's inequality.

The following fact is useful in the proof.

**Fact 1.** $e^x \leq 1 + x + 2x^2$ for any $|x| \leq 1$.

*Proof.* By Taylor's expansion, we known for any $x \in \mathbb{R}$, there exists a point $\hat{x}$ in between $0$ and $x$ such that $e^x = 1 + x + e^{\hat{x}}\frac{x^2}{2}$. (Note that the value of $\hat{x}$ depends on $x$ and if $x > 0$, then $\hat{x} \in (0, x)$; if $x < 0$, then $\hat{x} \in (x, 0)$; and if $x = 0$, then $\hat{x} = x$.) Since $|x| \leq 1$, we have $e^{\hat{x}} \leq e \leq 4$. Thus, $e^x \leq 1 + x + 2x^2$ for any $|x| \leq 1$. $\square$

The next lemma provides an upper bound of $\mathbb{E}[e^{rZ(t)}]$ with constant $r = \frac{\zeta}{4t_0\delta_{\max}^2} < 1$.

**Lemma 10.** *Under the assumption of Lemma 5, we have*

$$\mathbb{E}[e^{rZ(t)}] \leq \frac{e^{rt_0\delta_{\max}}}{1 - \rho} e^{r\theta}, \forall t \in \{0, 1, \ldots\},$$

*where* $r = \frac{\zeta}{4t_0\delta_{\max}^2}$, $\rho = 1 - \frac{\zeta^2}{8\delta_{\max}^2} = 1 - \frac{rt_0\zeta}{2}$.

*Proof.* Since $0 < \zeta < \delta_{\max}$, we have $0 < \rho < 1 < e^{r\delta_{\max}}$. Define $\eta(t) = Z(t+t_0) - Z(t)$. Note that $|\eta(t)| \leq t_0\delta_{\max}, \forall t \geq 0$ and $|r\eta(t)| \leq \frac{\zeta}{4t_0\delta_{\max}^2}t_0\delta_{\max} = \frac{\zeta}{4\delta_{\max}} \leq 1$. Then,

$$e^{rZ(t+t_0)} = e^{rZ(t)}e^{r\eta(t)} \tag{16}$$

$$\overset{(a)}{\leq} e^{rZ(t)}[1 + r\eta(t) + 2r^2t_0^2\delta_{\max}^2]$$

$$\overset{(b)}{=} e^{rZ(t)}[1 + r\eta(t) + \frac{1}{2}rt_0\zeta], \tag{17}$$

where (a) follows from Fact 1 by noting that $|r\eta(t)| \leq 1$ and $|\eta(t)| \leq t_0\delta_{\max}$; and (b) follows by substituting $r = \frac{\zeta}{4t_0\delta_{\max}^2}$ into a single $r$ of the term $2r^2t_0^2\delta_{\max}^2$.

Next, consider the cases $Z(t) \geq \theta$ and $Z(t) < \theta$, separately.

- Case $Z(t) \geq \theta$: Taking conditional expectations on both sides of (17) yields:

$$\mathbb{E}[e^{rZ(t+t_0)}|Z(t)] \leq \mathbb{E}[e^{rZ(t)}(1 + r\eta(t) + \frac{1}{2}rt_0\zeta)|Z(t)]$$

$$\overset{(a)}{\leq} e^{rZ(t)}[1 - rt_0\zeta + \frac{1}{2}rt_0\zeta]$$

$$= e^{rZ(t)}[1 - \frac{rt_0\zeta}{2}]$$

$$\overset{(b)}{=} \rho e^{rZ(t)}.$$

  where (a) follows from the fact that $\mathbb{E}[Z(t+t_0) - Z(t)|\mathcal{F}(t)] \leq -t_0\zeta$ when $Z(t) \geq \theta$; and (b) follows from the fact that $\rho = 1 - \frac{rt_0\zeta}{2}$.

- Case $Z(t) < \theta$: Taking conditional expectations on both sides of (16) yields:

$$\mathbb{E}[e^{rZ(t+t_0)}|Z(t)] = \mathbb{E}[e^{rZ(t)}e^{r\eta(t)}|Z(t)]$$

$$= e^{rZ(t)}\mathbb{E}[e^{r\eta(t)}|Z(t)]$$

$$\overset{(a)}{\leq} e^{rt_0\delta_{\max}}e^{rZ(t)},$$

  where (a) follows from the fact that $\eta(t) \leq t_0\delta_{\max}$.

Putting two cases together yields:

$$\mathbb{E}[e^{rZ(t+t_0)}] \overset{(a)}{=} \Pr(Z(t) \geq \theta)\mathbb{E}[e^{rZ(t+t_0)}|Z(t) \geq \theta] + \Pr(Z(t) < \theta)\mathbb{E}[e^{rZ(t+t_0)}|Z(t) < \theta]$$

$$\overset{(b)}{\leq} \rho\mathbb{E}[e^{rZ(t)}|Z(t) \geq \theta]\Pr(Z(t) \geq \theta) + e^{rt_0\delta_{\max}}\mathbb{E}[e^{rZ(t)}|Z(t) < \theta]\Pr(Z(t) < \theta)$$

$$\overset{(c)}{=} \rho\mathbb{E}[e^{rZ(t)}] + [e^{rt_0\delta_{\max}} - \rho]\mathbb{E}[e^{rZ(t)}|Z(t) < \theta]\Pr(Z(t) < \theta)$$

$$\overset{(d)}{\leq} \rho\mathbb{E}[e^{rZ(t)}] + [e^{rt_0\delta_{\max}} - \rho]e^{r\theta}$$

$$\leq \rho\mathbb{E}[e^{rZ(t)}] + e^{rt_0\delta_{\max}}e^{r\theta}, \tag{18}$$

where (a) follows by the definition of expectations; (b) follows from the results in the above two cases; (c) follows from the fact that $\mathbb{E}[e^{rZ(t)}] = \Pr(Z(t) \geq \theta)\mathbb{E}[e^{rZ(t)}|Z(t) \geq \theta] + \Pr(Z(t) < \theta)\mathbb{E}[e^{rZ(t)}|Z(t) < \theta]$; and (d) follow from the fact that $e^{rt_0\delta_{\max}} > \rho$.

Now, we prove $\mathbb{E}[e^{rZ(t)}] \leq \frac{e^{rt_0\delta_{\max}}}{1-\rho}e^{r\theta}, \forall t \geq 0$, by inductions.

We first consider the base case $t \in \{0, 1, \ldots, t_0\}$. Since $Z(t) \leq t\delta_{\max}, \forall t \geq 0$, it follows that $\mathbb{E}[e^{rZ(t)}] \leq e^{rt\delta_{\max}} \leq e^{rt_0\delta_{\max}} \leq \frac{e^{rt_0\delta_{\max}}}{1-\rho}e^{r\theta}, \forall t \in \{0, 1, \ldots, t_0\}$, where the last inequality follows because $\frac{e^{r\theta}}{1-\rho} \geq 1$.

Now assume that $\mathbb{E}[e^{rZ(t)}] \leq \frac{e^{rt_0\delta_{\max}}}{1-\rho}e^{r\theta}$ for all $t \in \{0, 1, \ldots, \tau\}$ with some $\tau \geq t_0$ and consider iteration $t = \tau + 1$. By (18), we have

$$\begin{aligned}
\mathbb{E}[e^{rZ(\tau+1)}] \leq& \rho\mathbb{E}[e^{rZ(\tau+1-t_0)}] + e^{rt_0\delta_{\max}}e^{r\theta} \\
\overset{(a)}{\leq}& \rho\frac{e^{rt_0\delta_{\max}}}{1-\rho}e^{r\theta} + e^{rt_0\delta_{\max}}e^{r\theta} \\
=& \frac{e^{rt_0\delta_{\max}}}{1-\rho}e^{r\theta}
\end{aligned}$$

where (a) follows from the induction hypothesis by noting that $0 \leq \tau + 1 - t_0 \leq \tau$.

Thus, this lemma follows by inductions. □

By this lemma, for all $t \in \{0, 1, \ldots\}$, we have

$$\mathbb{E}[e^{rZ(t)}] \leq \frac{e^{rt_0\delta_{\max}}}{1-\rho}e^{r\theta}. \tag{19}$$

**Proof of Part (1):** Note that $e^{rx}$ is convex with respect to $x$ when $r > 0$. By Jensen's inequality,

$$\begin{aligned}
e^{r\mathbb{E}[Z(t)]} \leq& \mathbb{E}[e^{rZ(t)}] \\
\overset{(a)}{\leq}& \frac{e^{r(\theta+t_0\delta_{\max})}}{1-\rho},
\end{aligned} \tag{20}$$

where (a) follows from (19).

Taking logarithm on both sides and dividing by $r$ yields:

$$\begin{aligned}
\mathbb{E}[Z(t)] \leq& \theta + t_0\delta_{\max} + \frac{1}{r}\log\Big[\frac{1}{1-\rho}\Big] \\
\overset{(a)}{=}& \theta + t_0\delta_{\max} + t_0\frac{4\delta_{\max}^2}{\zeta}\log\Big[\frac{8\delta_{\max}^2}{\zeta^2}\Big],
\end{aligned}$$

where (a) follows by recalling that $r = \frac{\zeta}{4t_0\delta_{\max}^2}$ and $\rho = 1 - \frac{\zeta^2}{8\delta_{\max}^2}$.

**Proof of Part (2):** Fix $z$. Note that

$$\begin{aligned}
\Pr(Z(t) \geq z) =& \Pr(e^{rZ(t)} \geq e^{rz}) \\
\overset{(a)}{\leq}& \frac{\mathbb{E}[e^{rZ(t)}]}{e^{rz}} \\
\overset{(b)}{\leq}& e^{r(\theta-z+t_0\delta_{\max})}\frac{1}{1-\rho} \\
\overset{(c)}{=}& e^{\frac{\zeta}{4t_0\delta_{\max}^2}(\theta-z+t_0\delta_{\max})}\Big[\frac{8\delta_{\max}^2}{\zeta^2}\Big]
\end{aligned} \tag{21}$$

where (a) follows from Markov's inequality; (b) follows from (19); and (c) follows by recalling that $r = \frac{\zeta}{4t_0\delta_{\max}^2}$ and $\rho = 1 - \frac{\zeta^2}{8\delta_{\max}^2}$.

Define $\mu = e^{\frac{\zeta}{4t_0\delta_{\max}^2}(\theta-z+t_0\delta_{\max})}\Big[\frac{8\delta_{\max}^2}{\zeta^2}\Big]$. It follows that if

$$z = \theta + t_0\delta_{\max} + t_0\frac{4\delta_{\max}^2}{\zeta}\log\Big[\frac{8\delta_{\max}^2}{\zeta^2}\Big] + t_0\frac{4\delta_{\max}^2}{\zeta}\log(\frac{1}{\mu}),$$

then we have $\Pr(Z(t) \geq z) \leq \mu$ by (21).

## 7.5 Proof of Lemma 7

The next lemma will be useful in our proof.

**Lemma 11.** *Let $\hat{\mathbf{x}} \in \mathcal{X}_0$ be a Slater point defined in Assumption 2, i.e, $\tilde{g}_k(\hat{\mathbf{x}}) = \mathbb{E}_\omega[g_k(\hat{\mathbf{x}}; \omega)] \leq -\epsilon, \forall k \in \{1, 2, \ldots, m\}$. Then*

$$\mathbb{E}[\sum_{k=1}^m Q_k(t_1)g_k^{t_1}(\hat{\mathbf{x}})|\mathcal{W}(t_2)] \leq -\epsilon\mathbb{E}[\|\mathbf{Q}(t_1)\||\mathcal{W}(t_2)], \quad \forall t_2 \leq t_1 - 1$$

*where $\epsilon > 0$ is defined in Assumption 2.*

*Proof.* To prove this lemma, we first show that

$$\mathbb{E}[Q_k(t_1)g_k^{t_1}(\hat{\mathbf{x}})|\mathcal{W}(t_2)] \leq -\epsilon\mathbb{E}[Q_k(t_1)|\mathcal{W}(t_2)], \forall k \in \{1, 2, \ldots, m\}, \forall t_2 \leq t_1 - 1.$$

Fix $k \in \{1, 2, \ldots, m\}$. Note that $\mathbf{Q}(t_1) \in \mathcal{W}(t_1 - 1)$ and $g_k^{t_1}(\hat{\mathbf{x}})$ is independent of $\mathcal{W}(t_1 - 1)$. Further, if $t_2 \leq t_1 - 1$, then $\mathcal{W}(t_2) \subseteq \mathcal{W}(t_1 - 1)$. Thus, we have

$$\mathbb{E}[Q_k(t_1)g_k^{t_1}(\hat{\mathbf{x}})|\mathcal{W}(t_2)] \overset{(a)}{=} \mathbb{E}\big[\mathbb{E}[Q_k(t_1)g_k^{t_1}(\hat{\mathbf{x}})|\mathcal{W}(t_1 - 1)]|\mathcal{W}(t_2)\big]$$

$$\overset{(b)}{=} \mathbb{E}\big[Q_k(t_1)\mathbb{E}[g_k^{t_1}(\hat{\mathbf{x}})]|\mathcal{W}(t_2)\big]$$

$$\overset{(c)}{=} \mathbb{E}[g_k^{t_1}(\hat{\mathbf{x}})]\mathbb{E}[Q_k(t_1)|\mathcal{W}(t_2)]$$

$$\overset{(d)}{\leq} -\epsilon\mathbb{E}[Q_k(t_1)|\mathcal{W}(t_2)]$$

where (a) follows from iterated expectations; (b) follows because $g_k^{t_1}(\hat{\mathbf{x}})$ is independent of $\mathcal{W}(t_1 - 1)$ and $Q_k(t_1) \in \mathcal{W}(t_1 - 1)$; (c) follows by extracting the constant $\mathbb{E}[g_k^{t_1}(\hat{\mathbf{x}})]$ and (d) follows from the assumption that $\hat{\mathbf{x}}$ is a Slater point, $g^t(\cdot)$ are i.i.d. across $t$ and the fact that $Q_k(t) \geq 0$.

Now, summing over $m \in \{1, 2, \ldots, m\}$ yields

$$\mathbb{E}[\sum_{k=1}^m Q_k(t_1)g_k^{t_1}(\hat{\mathbf{x}})|\mathcal{W}(t_2)] \leq -\epsilon\mathbb{E}[\sum_{k=1}^m Q_k(t_1)|\mathcal{W}(t_2)]$$

$$\overset{(a)}{\leq} -\epsilon\mathbb{E}[\|\mathbf{Q}(t_1)\||\mathcal{W}(t_2)]$$

where (a) follows from the basic fact that $\sum_{k=1}^m a_k \geq \sqrt{\sum_{k=1}^m a_k^2}$ when $a_k \geq 0, \forall k \in \{1, 2, \ldots, m\}$. $\square$

The bounded difference of $|\mathbf{Q}(t+1) - \mathbf{Q}(t)|$ follows directly from the virtual queue update equation (3) and is summarized in the next Lemma.

**Lemma 12.** *Let $\mathbf{Q}(t), t \in \{0, 1, \ldots\}$ be the sequence generated by Algorithm 1. Then,*

$$\|\mathbf{Q}(t)\| - G - \sqrt{m}D_2R \leq \|\mathbf{Q}(t+1)\| \leq \|\mathbf{Q}(t)\| + G, \forall t \geq 0.$$

*Proof.*

- **Proof of $\|\mathbf{Q}(t+1)\| \leq \|\mathbf{Q}(t)\| + G$:**

  Fix $t \geq 0$ and $k \in \{1, 2, \ldots, m\}$. The virtual queue update equation implies that

  $$Q_k(t+1) = \max\{Q_k(t) + g_k^t(\mathbf{x}(t)) + [\nabla g_k^t(\mathbf{x}(t))]^\mathsf{T}[\mathbf{x}(t+1) - \mathbf{x}(t)], 0\}$$

  $$\overset{(a)}{\leq} \max\{Q_k(t) + g_k^t(\mathbf{x}(t+1)), 0\},$$

  where (a) follows from the convexity of $g_k^t(\cdot)$.

  Note that $Q_k(t+1) \geq 0$ and recall the fact that if $0 \leq a \leq \max\{b, 0\}$, then $a^2 \leq b^2$ for all $a, b \in \mathbb{R}$. Then, we have $[Q_k(t+1)]^2 \leq [Q_k(t) + g_k^t(\mathbf{x}(t+1))]^2$.

  Summing over $k \in \{1, 2, \ldots, m\}$ yields

  $$\|\mathbf{Q}(t+1)\|^2 \leq \|\mathbf{Q}(t) + \mathbf{g}^t(\mathbf{x}(t+1))\|^2.$$

  Thus, $\|\mathbf{Q}(t+1)\| \leq \|\mathbf{Q}(t) + \mathbf{g}^t(\mathbf{x}(t+1))\| \leq \|\mathbf{Q}(t)\| + \|\mathbf{g}^t(\mathbf{x}(t+1))\| \leq \|\mathbf{Q}(t)\| + G$ where the last inequality follows from Assumption 1.

- **Proof of** $\|\mathbf{Q}(t+1)\| \geq \|\mathbf{Q}(t)\| - G - \sqrt{m}D_2R$:

  Since $Q_k(t) \geq 0$, it follows that $|Q_k(t+1) - Q_k(t)| \leq |g_k^t(\mathbf{x}(t)) + [\nabla g_k^t(\mathbf{x}(t))]^{\mathsf{T}}[\mathbf{x}(t+1) - \mathbf{x}(t)]|$. (This can be shown by considering $g_k^t(\mathbf{x}(t)) + [\nabla g_k^t(\mathbf{x}(t))]^{\mathsf{T}}[\mathbf{x}(t+1) - \mathbf{x}(t)] \geq 0$ and $g_k^t(\mathbf{x}(t)) + [\nabla g_k^t(\mathbf{x}(t))]^{\mathsf{T}}[\mathbf{x}(t+1) - \mathbf{x}(t)] < 0$ separately.) Thus, we have $\|\mathbf{Q}(t+1) - \mathbf{Q}(t)\| \leq G + \sqrt{m}D_2R$, which further implies $\|\mathbf{Q}(t+1)\| \geq \|\mathbf{Q}(t)\| - G - \sqrt{m}D_2R$ by the triangle inequality of norms.

  $\square$

Now, we are ready to present the main proof of Lemma 7. Note that Lemma 12 gives $\big|\|\mathbf{Q}(t+1)\| - \|\mathbf{Q}(t)\|\big| \leq G + \sqrt{m}D_2R$, which further implies that $\mathbb{E}[\|\mathbf{Q}(t+t_0)\| - \|\mathbf{Q}(t)\| \,\big|\, \mathbf{Q}(t)] \leq t_0(G + \sqrt{m}D_2R)$ when $\|\mathbf{Q}(t)\| < \theta$. It remains to prove $\mathbb{E}[\|\mathbf{Q}(t+1)\| - \|\mathbf{Q}(t)\| \,\big|\, \mathbf{Q}(t)] \leq -\frac{\epsilon}{2}t_0$ when $\|\mathbf{Q}(t)\| \geq \theta$. Note that $\|\mathbf{Q}(0)\| = 0 < \theta$.

Fix $t \geq 1$ and consider that $\|\mathbf{Q}(t)\| \geq \theta$. Let $\hat{\mathbf{x}} \in \mathcal{X}_0$ and $\epsilon > 0$ be defined in Assumption 2. Note that $\mathbb{E}[g_k^t(\hat{\mathbf{x}})] \leq -\epsilon, \forall k \in \{1, 2, \dots, m\}, \forall t \in \{1, 2, \dots\}$ since $\omega(t)$ are i.i.d. from the distribution of $\omega$. Since $\hat{\mathbf{x}} \in \mathcal{X}_0$, by Lemma 4, for all $\tau \in \{t, t+1, \dots, t+t_0-1\}$, we have

$$V[\nabla f^\tau(\mathbf{x}(\tau))]^{\mathsf{T}}[\mathbf{x}(\tau+1) - \mathbf{x}(\tau)] + \sum_{k=1}^m Q_k(\tau)[\nabla g_k^\tau(\mathbf{x}(\tau))]^{\mathsf{T}}[\mathbf{x}(\tau+1) - \mathbf{x}(\tau)] + \alpha\|\mathbf{x}(\tau+1) - \mathbf{x}(\tau)\|^2$$

$$\leq V[\nabla f^\tau(\mathbf{x}(\tau))]^{\mathsf{T}}[\hat{\mathbf{x}} - \mathbf{x}(\tau)] + \sum_{k=1}^m Q_k(\tau)[\nabla g_k^\tau(\mathbf{x}(\tau))]^{\mathsf{T}}[\hat{\mathbf{x}} - \mathbf{x}(\tau)] + \alpha[\|\hat{\mathbf{x}} - \mathbf{x}(\tau)\|^2 - \|\hat{\mathbf{x}} - \mathbf{x}(\tau+1)\|^2].$$

Adding $\sum_{k=1}^m Q_k(\tau)g_k^\tau(\mathbf{x}(\tau))$ on both sides and noting that $g_k^\tau(\mathbf{x}(\tau)) + [\nabla g_k^\tau(\mathbf{x}(\tau))]^{\mathsf{T}}[\hat{\mathbf{x}} - \mathbf{x}(\tau)] \leq g_k^\tau(\hat{\mathbf{x}})$ by convexity yields

$$V[\nabla f^\tau(\mathbf{x}(\tau))]^{\mathsf{T}}[\mathbf{x}(\tau+1) - \mathbf{x}(\tau)] + \sum_{k=1}^m Q_k(\tau)\big[g_k^\tau(\mathbf{x}(\tau)) + [\nabla g_k^\tau(\mathbf{x}(\tau))]^{\mathsf{T}}[\mathbf{x}(\tau+1) - \mathbf{x}(\tau)]\big]$$

$$+ \alpha\|\mathbf{x}(\tau+1) - \mathbf{x}(\tau)\|^2$$

$$\leq V[\nabla f^\tau(\mathbf{x}(\tau))]^{\mathsf{T}}[\hat{\mathbf{x}} - \mathbf{x}(\tau)] + \sum_{k=1}^m Q_k(\tau)g_k^\tau(\hat{\mathbf{x}}) + \alpha[\|\hat{\mathbf{x}} - \mathbf{x}(\tau)\|^2 - \|\hat{\mathbf{x}} - \mathbf{x}(\tau+1)\|^2].$$

Rearranging terms yields

$$\sum_{k=1}^m Q_k(t)\big[g_k^\tau(\mathbf{x}(t)) + [\nabla g_k^\tau(\mathbf{x}(\tau))]^{\mathsf{T}}[\mathbf{x}(\tau+1) - \mathbf{x}(\tau)]\big]$$

$$\leq V[\nabla f^\tau(\mathbf{x}(\tau))]^{\mathsf{T}}[\hat{\mathbf{x}} - \mathbf{x}(\tau)] - V[\nabla f^\tau(\mathbf{x}(\tau))]^{\mathsf{T}}[\mathbf{x}(\tau+1) - \mathbf{x}(\tau)]$$

$$+ \alpha[\|\hat{\mathbf{x}} - \mathbf{x}(\tau)\|^2 - \|\hat{\mathbf{x}} - \mathbf{x}(\tau+1)\|^2] - \alpha\|\mathbf{x}(\tau+1) - \mathbf{x}(\tau)\|^2 + \sum_{k=1}^m Q_k(t)g_k^\tau(\hat{\mathbf{x}})$$

$$\leq V[\nabla f^\tau(\mathbf{x}(\tau))]^{\mathsf{T}}[\hat{\mathbf{x}} - \mathbf{x}(\tau+1)] + \alpha[\|\hat{\mathbf{x}} - \mathbf{x}(\tau)\|^2 - \|\hat{\mathbf{x}} - \mathbf{x}(\tau+1)\|^2] + \sum_{k=1}^m Q_k(\tau)g_k^\tau(\hat{\mathbf{x}})$$

$$\overset{(a)}{\leq} V\|\nabla f^\tau(\mathbf{x}(\tau))\|\|\hat{\mathbf{x}} - \mathbf{x}(\tau+1)\| + \alpha[\|\hat{\mathbf{x}} - \mathbf{x}(\tau)\|^2 - \|\hat{\mathbf{x}} - \mathbf{x}(\tau+1)\|^2] + \sum_{k=1}^m Q_k(\tau)g_k^\tau(\hat{\mathbf{x}})$$

$$\overset{(b)}{\leq} VD_1R + \alpha[\|\hat{\mathbf{x}} - \mathbf{x}(\tau)\|^2 - \|\hat{\mathbf{x}} - \mathbf{x}(\tau+1)\|^2] + \sum_{k=1}^m Q_k(\tau)g_k^\tau(\hat{\mathbf{x}}), \tag{22}$$

where (a) follows from the Cauchy-Schwarz inequality and (b) follows from Assumption 1.

By Lemma 2, for all $\tau \in \{t, t+1, \dots, t+t_0-1\}$, we have

$$\Delta(\tau) \leq \sum_{k=1}^m Q_k(\tau)\big[g_k^\tau(\mathbf{x}(\tau)) + [\nabla g_k^\tau(\mathbf{x}(\tau))]^{\mathsf{T}}[\mathbf{x}(\tau+1) - \mathbf{x}(\tau)]\big] + \frac{1}{2}[G + \sqrt{m}D_2R]^2$$

$$\overset{(a)}{\leq} VD_1R + \frac{1}{2}[G + \sqrt{m}D_2R]^2 + \alpha[\|\hat{\mathbf{x}} - \mathbf{x}(\tau)\|^2 - \|\hat{\mathbf{x}} - \mathbf{x}(\tau+1)\|^2] + \sum_{k=1}^m Q_k(\tau)g_k^\tau(\hat{\mathbf{x}}),$$

where (a) follows from (22).

Summing the above inequality over $\tau \in \{t, t+1, \ldots, t+t_0-1\}$, taking expectations conditional on $\mathcal{W}(t-1)$ on both sides and recalling that $\Delta(\tau) = \frac{1}{2}\|\mathbf{Q}(\tau+1)\|^2 - \frac{1}{2}\|\mathbf{Q}(\tau)\|^2$ yields

$$
\mathbb{E}[\|\mathbf{Q}(t+t_0)\|^2 - \|\mathbf{Q}(t)\|^2 | \mathcal{W}(t-1)]
$$
$$
\leq 2VD_1Rt_0 + t_0[G + \sqrt{m}D_2R]^2 + 2\alpha\mathbb{E}[\|\hat{\mathbf{x}} - \mathbf{x}(t)\|^2 - \|\hat{\mathbf{x}} - \mathbf{x}(t+t_0)\|^2 | \mathcal{W}(t-1)]
$$
$$
+ 2\sum_{\tau=t}^{t+t_0-1}\mathbb{E}[\sum_{k=1}^{m}Q_k(\tau)g_k^\tau(\hat{\mathbf{x}})|\mathcal{W}(t-1)]
$$
$$
\stackrel{(a)}{\leq} 2VD_1Rt_0 + t_0[G + \sqrt{m}D_2R]^2 + 2\alpha R^2 - 2\epsilon\sum_{\tau=t}^{t+t_0-1}\mathbb{E}[\|\mathbf{Q}(\tau)\| | \mathcal{W}(t-1)]
$$
$$
\stackrel{(b)}{\leq} 2VD_1Rt_0 + t_0[G + \sqrt{m}D_2R]^2 + 2\alpha R^2 - 2\epsilon\sum_{\tau=0}^{t_0-1}\mathbb{E}[\|\mathbf{Q}(t)\| - \tau(G + \sqrt{m}D_2R)|\mathcal{W}(t-1)]
$$
$$
= 2VD_1Rt_0 + t_0[G + \sqrt{m}D_2R]^2 + 2\alpha R^2 - 2\epsilon t_0\|\mathbf{Q}(t)\| + \epsilon t_0(t_0-1)(G + \sqrt{m}D_2R)
$$
$$
\leq 2VD_1Rt_0 + t_0[G + \sqrt{m}D_2R]^2 + 2\alpha R^2 - 2\epsilon t_0\|\mathbf{Q}(t)\| + \epsilon t_0^2(G + \sqrt{m}D_2R)
$$

where (a) follows from $\|\hat{\mathbf{x}} - \mathbf{x}(t)\|^2 - \|\hat{\mathbf{x}} - \mathbf{x}(t+t_0)\|^2 \leq R^2$ by Assumption 1 and $\mathbb{E}[\sum_{k=1}^{m}Q_k(\tau)g_k^\tau(\hat{\mathbf{x}})|\mathcal{W}(t-1)] \leq -\epsilon\mathbb{E}[\|\mathbf{Q}(\tau)\| | \mathcal{W}(t-1)], \forall\tau \in \{t, t+1, \ldots, t+t_0-1\}$ by Lemma 11; (b) follows from $\|\mathbf{Q}(t+1)\| \geq \|\mathbf{Q}(t)\| - (G + \sqrt{m}D_2R), \forall t$ by Lemma 12.

This inequality can be rewritten as

$$
\mathbb{E}[\|\mathbf{Q}(t+t_0)\|^2 | \mathcal{W}(t-1)]
$$
$$
\leq \|\mathbf{Q}(t)\|^2 - 2\epsilon t_0\|\mathbf{Q}(t)\| + 2VD_1Rt_0 + 2\alpha R^2 + t_0[G + \sqrt{m}D_2R]^2 + \epsilon t_0^2(G + \sqrt{m}D_2R)
$$
$$
\stackrel{(a)}{\leq} \|\mathbf{Q}(t)\|^2 - \epsilon t_0\|\mathbf{Q}(t)\| - \epsilon t_0[\frac{\epsilon}{2}t_0 + (G + \sqrt{m}D_2R)t_0 + \frac{2\alpha R^2}{t_0\epsilon} + \frac{2VD_1R + [G + \sqrt{m}D_2R]^2}{\epsilon}]
$$
$$
+ 2VD_1Rt_0 + 2\alpha R^2 + t_0[G + \sqrt{m}D_2R]^2 + \epsilon t_0^2(G + \sqrt{m}D_2R)
$$
$$
= \|\mathbf{Q}(t)\|^2 - \epsilon t_0\|\mathbf{Q}(t)\| - \frac{\epsilon^2 t_0^2}{2}
$$
$$
\leq [\|\mathbf{Q}(t)\| - \frac{\epsilon}{2}t_0]^2,
$$

where (a) follows from the hypothesis that $\|\mathbf{Q}(t)\| \geq \theta = \frac{\epsilon}{2}t_0 + (G + \sqrt{m}D_2R)t_0 + \frac{2\alpha R^2}{t_0\epsilon} + \frac{2VD_1R + [G + \sqrt{m}D_2R]^2}{\epsilon}$.

Taking square root on both sides yields

$$
\sqrt{\mathbb{E}[\|\mathbf{Q}(t+t_0)\|^2 | \mathcal{W}(t-1)]} \leq \|\mathbf{Q}(t)\| - \frac{\epsilon}{2}t_0.
$$

By the concavity of function $\sqrt{x}$ and Jensen's inequality, we have

$$
\mathbb{E}[\|\mathbf{Q}(t+t_0)\| | \mathcal{W}(t-1)] \leq \sqrt{\mathbb{E}[\|\mathbf{Q}(t+t_0)\|^2 | \mathcal{W}(t-1)]} \leq \|\mathbf{Q}(t)\| - \frac{\epsilon}{2}t_0.
$$

## 7.6 Proof of Lemma 8

Fix $t \geq 1$. By Lemma 4, we have

$$
V[\nabla f^t(\mathbf{x}(t))]^\mathsf{T}[\mathbf{x}(t+1) - \mathbf{x}(t)] + \sum_{k=1}^{m}Q_k(t)[\nabla g_k^t(\mathbf{x}(t))]^\mathsf{T}[\mathbf{x}(t+1) - \mathbf{x}(t)] + \alpha\|\mathbf{x}(t+1) - \mathbf{x}(t)\|^2
$$
$$
\leq V[\nabla f^t(\mathbf{x}(t))]^\mathsf{T}[\mathbf{z} - \mathbf{x}(t)] + \sum_{k=1}^{m}Q_k(t)[\nabla g_k^t(\mathbf{x}(t))]^\mathsf{T}[\mathbf{z} - \mathbf{x}(t)] + \alpha[\|\mathbf{z} - \mathbf{x}(t)\|^2 - \|\mathbf{z} - \mathbf{x}(t+1)\|^2].
$$

Adding constant $Vf^t(\mathbf{x}(t)) + \sum_{k=1}^{m} Q_k(t)g_k^t(\mathbf{x}(t))$ on both sides; and noting that $f^t(\mathbf{x}(t)) + [\nabla f^t(\mathbf{x}(t))]^\mathsf{T}[\mathbf{z} - \mathbf{x}(t)] \leq f^t(\mathbf{z})$ and $g_k^t(\mathbf{x}(t)) + [\nabla g_k^t(\mathbf{x}(t))]^\mathsf{T}[\mathbf{z} - \mathbf{x}(t)] \leq g_k^t(\mathbf{z})$ by convexity yields

$$Vf^t(\mathbf{x}(t)) + V[\nabla f^t(\mathbf{x}(t))]^\mathsf{T}[\mathbf{x}(t+1) - \mathbf{x}(t)] + \sum_{k=1}^{m} Q_k(t)\big[g_k^t(\mathbf{x}(t)) + [\nabla g_k^t(\mathbf{x}(t))]^\mathsf{T}[\mathbf{x}(t+1) - \mathbf{x}(t)]\big]$$

$$+ \alpha\|\mathbf{x}(t+1) - \mathbf{x}(t)\|^2$$

$$\leq Vf^t(\mathbf{z}) + \sum_{k=1}^{m} Q_k(t)g_k^t(\mathbf{z}) + \alpha[\|\mathbf{z} - \mathbf{x}(t)\|^2 - \|\mathbf{z} - \mathbf{x}(t+1)\|^2]. \tag{23}$$

By Lemma 2, we have

$$\Delta(t) \leq \sum_{k=1}^{m} Q_k(t)\big[g_k^t(\mathbf{x}(t)) + [\nabla g_k^t(\mathbf{x}(t))]^\mathsf{T}[\mathbf{x}(t+1) - \mathbf{x}(t)]\big] + \frac{1}{2}[G + \sqrt{m}D_2 R]^2. \tag{24}$$

Summing (23) and (24), cancelling common terms and rearranging terms yields

$$Vf^t(\mathbf{x}(t)) \leq Vf^t(\mathbf{z}) - \Delta(t) + \sum_{k=1}^{m} Q_k(t)g_k^t(\mathbf{z}) + \alpha[\|\mathbf{z} - \mathbf{x}(t)\|^2 - \|\mathbf{z} - \mathbf{x}(t+1)\|^2]$$

$$- V[\nabla f^t(\mathbf{x}(t))]^\mathsf{T}[\mathbf{x}(t+1) - \mathbf{x}(t)] - \alpha\|\mathbf{x}(t+1) - \mathbf{x}(t)\|^2 + \frac{1}{2}[G + \sqrt{m}D_2 R]^2 \tag{25}$$

Note that

$$- V[\nabla f^t(\mathbf{x}(t))]^\mathsf{T}[\mathbf{x}(t+1) - \mathbf{x}(t)] - \alpha\|\mathbf{x}(t+1) - \mathbf{x}(t)\|^2$$

$$\overset{(a)}{\leq} V\|\nabla f^t(\mathbf{x}(t))\|\|\mathbf{x}(t+1) - \mathbf{x}(t)\| - \alpha\|\mathbf{x}(t+1) - \mathbf{x}(t)\|^2$$

$$\overset{(b)}{\leq} V D_1\|\mathbf{x}(t+1) - \mathbf{x}(t)\| - \alpha\|\mathbf{x}(t+1) - \mathbf{x}(t)\|^2$$

$$= - \alpha\big[\|\mathbf{x}(t+1) - \mathbf{x}(t)\| - \frac{V D_1}{2\alpha}\big]^2 + \frac{V^2 D_1^2}{4\alpha}$$

$$\leq \frac{V^2 D_1^2}{4\alpha} \tag{26}$$

where (a) follows from the Cauchy-Schwarz inequality; and (b) follows from Assumption 1.

Substituting (26) into (25) yields

$$Vf^t(\mathbf{x}(t)) \leq Vf^t(\mathbf{z}) - \Delta(t) + \sum_{k=1}^{m} Q_k(t)g_k^t(\mathbf{z}) + \alpha[\|\mathbf{z} - \mathbf{x}(t)\|^2 - \|\mathbf{z} - \mathbf{x}(t+1)\|^2] + \frac{V^2 D_1^2}{4\alpha}$$

$$+ \frac{1}{2}[G + \sqrt{m}D_2 R]^2.$$

Summing over $t \in \{1, 2, \ldots, T\}$ yields

$$V\sum_{t=1}^{T} f^t(\mathbf{x}(t)) \leq V\sum_{t=1}^{T} f^t(\mathbf{z}) - \sum_{t=1}^{T}\Delta(t) + \alpha\sum_{t=1}^{T}[\|\mathbf{z} - \mathbf{x}(t)\|^2 - \|\mathbf{z} - \mathbf{x}(t+1)\|^2] + \frac{V^2 D_1^2}{4\alpha}T$$

$$+ \frac{1}{2}[G + \sqrt{m}D_2 R]^2 T + \sum_{t=1}^{T}\Big[\sum_{k=1}^{m} Q_k(t)g_k^t(\mathbf{z})\Big]$$

$$\overset{(a)}{=} V\sum_{t=1}^{T} f^t(\mathbf{z}) + L(1) - L(T+1) + \alpha\|\mathbf{z} - \mathbf{x}(1)\|^2 - \alpha\|\mathbf{z} - \mathbf{x}(T+1)\|^2 + \frac{V^2 D_1^2}{4\alpha}T$$

$$+ \frac{1}{2}[G + \sqrt{m}D_2 R]^2 T + \sum_{t=1}^{T}\Big[\sum_{k=1}^{m} Q_k(t)g_k^t(\mathbf{z})\Big]$$

$$\overset{(b)}{\leq} V\sum_{t=1}^{T} f^t(\mathbf{z}) + \alpha R^2 + \frac{V^2 D_1^2}{4\alpha}T + \frac{1}{2}[G + \sqrt{m}D_2 R]^2 T + \sum_{t=1}^{T}\Big[\sum_{k=1}^{m} Q_k(t)g_k^t(\mathbf{z})\Big].$$

where (a) follows by recalling that $\Delta(t) = L(t+1) - L(t)$; and (b) follows because $\|\mathbf{z} - \mathbf{x}(1)\| \leq R$ by Assumption 1, $L(1) = \frac{1}{2}\|\mathbf{Q}(1)\|^2 = 0$ and $L(T+1) = \frac{1}{2}\|\mathbf{Q}(T+1)\|^2 \geq 0$.

Dividing both sides by $V$ yields the desired result.

## 7.7 Proof of Theorem 3

Define random process $Z(t) = \|\mathbf{Q}(t)\|, \forall t \in \{1, 2, \ldots\}$. By Lemma 7, $Z(t)$ satisfies the conditions in Lemma 5 with $\delta_{\max} = G + \sqrt{m}D_2 R$, $\zeta = \frac{\epsilon}{2}$ and

$$\theta = \frac{\epsilon}{2}t_0 + (G + \sqrt{m}D_2 R)t_0 + \frac{2\alpha R^2}{t_0\epsilon} + \frac{2VD_1 R + [G + \sqrt{m}D_2 R]^2}{\epsilon}.$$

Fix $T \geq 1$ and $0 < \lambda < 1$. Taking $\mu = \lambda/(T+1)$ in part (2) of Lemma 5 yields

$$\Pr(\|\mathbf{Q}(t)\| \geq \gamma) \leq \frac{\lambda}{T+1}, \forall t \in \{1, 2, \ldots, T+1\},$$

where $\gamma = \frac{\epsilon}{2}t_0 + 2(G + \sqrt{m}D_2 R)t_0 + \frac{2\alpha R^2}{t_0\epsilon} + \frac{2VD_1 R + [G+\sqrt{m}D_2 R]^2}{\epsilon} + t_0 \frac{8[G+\sqrt{m}D_2 R]^2}{\epsilon} \log[\frac{32[G+\sqrt{m}D_2 R]^2}{\epsilon^2}] + t_0 \frac{8[G+\sqrt{m}D_2 R]^2}{\epsilon} \log(\frac{T+1}{\lambda})$.

By union bounds, we have

$$\Pr(\|\mathbf{Q}(t)\| \geq \gamma \text{ for some } t \in \{1, 2, \ldots, T+1\}) \leq \lambda.$$

This implies

$$\Pr(\|\mathbf{Q}(t)\| \leq \gamma \text{ for all } t \in \{1, 2, \ldots, T+1\}) \geq 1 - \lambda. \tag{27}$$

Taking $t_0 = \lceil \sqrt{T} \rceil$, $V = \sqrt{T}$ and $\alpha = T$ yields

$$\gamma = O(\sqrt{T}\log(T)) + O(\sqrt{T}\log(\frac{1}{\lambda})) = O(\sqrt{T}\log(T)\log(\frac{1}{\lambda})) \tag{28}$$

Recall that by Corollary 2 (with $V = \sqrt{T}$ and $\alpha = T$), for all $k \in \{1, 2, \ldots, m\}$, we have

$$\sum_{t=1}^{T} g_k(\mathbf{x}(t)) \leq \|\mathbf{Q}(T+1)\| + \frac{\sqrt{T}D_1 D_2}{2} + \frac{\sqrt{m}D_2^2}{2T} \sum_{t=1}^{T} \|\mathbf{Q}(t)\|. \tag{29}$$

It follows from (27)-(29) that

$$\Pr\left(\sum_{t=1}^{T} g_k(\mathbf{x}(t)) \leq O(\sqrt{T}\log(T)\log(\frac{1}{\lambda}))\right) \geq 1 - \lambda.$$

## 7.8 Proof of Lemma 9

Intuitively, the second term on the right side in the lemma bounds the probability that $|Z(\tau + 1) - Z(\tau)| > c$ for any $\tau \in \{0, 1, \ldots, t-1\}$, while the first term on the right side comes from the conventional Hoeffding-Azuma inequality. However, it is unclear whether or not $Z(t)$ is still a supermartigale conditional on the event that $|Z(\tau+1) - Z(\tau)| \leq c$ for any $\tau \in \{0, 1, \ldots, t-1\}$. That's why it is important to have $\{|Z(t+1) - Z(t)| > c\} \subseteq \{Y(t) > 0\}$ and $Y(t) \in \mathcal{F}(t)$, which means the boundedness of $|Z(t+1) - Z(t)|$ can be inferred from another random variable $Y(t)$ that belongs to $\mathcal{F}(t)$. The proof of Lemma 9 uses the truncation method to construct an auxiliary supermargingale.

Recall the definition of stoping time given as follows:

**Definition 1** ([7])**.** *Let* $\{\emptyset, \Omega\} = \mathcal{F}(0) \subseteq \mathcal{F}(1) \subseteq \mathcal{F}(2) \cdots$ *be a filtration. A discrete random variable* $T$ *is a stoping time (also known as an option time) if for any integer* $t < \infty$,

$$\{T = t\} \in \mathcal{F}(t),$$

*i.e. the event that the stopping time occurs at time* $t$ *is contained in the information up to time* $t$.

The next theorem summarizes that a supermartingale truncated at a stoping time is still a supermartingale.

**Theorem 5.** *(Theorem 5.2.6 in [7]) If random variable $T$ is a stopping time and $Z(t)$ is a supermartingale, then $Z(t \wedge T)$ is also a supermartingale, where $a \wedge b \triangleq \min\{a, b\}$.*

To prove this lemma, we first construct a new supermartingale by truncating the original supermartingale at a carefully chosen stopping time such that the new supermartingale has bounded differences.

Define integer random variable $T = \inf\{t \geq 0 : Y(t) > 0\}$. That is, $T$ is the first time $t$ when $Y(t) > 0$ happens. Now, we show that $T$ is a stoping time and if we define $\widetilde{Z}(t) = Z(t \wedge T)$, then $\{\widetilde{Z}(t) \neq Z(t)\} \subseteq \bigcup_{\tau=0}^{t-1}\{Y(\tau) > 0\}, \forall t \geq 1$ and $\widetilde{Z}(t)$ is a supermartingale with differences bounded by $c$.

1. **To show $T$ is a stoping time:** Note that $\{T = 0\} = \{Y(0) > 0\} \in \mathcal{F}(0)$. Fix integer $t' > 0$, we have

$$\{T = t'\} = \big\{ \inf\{t \geq 0 : Y(t) > 0\} = t' \big\}$$
$$= \big\{ \cap_{\tau=0}^{t'-1} \{|Y(\tau) \leq 0\} \big\} \cap \{Y(t') > 0\}$$
$$\overset{(a)}{\in} \mathcal{F}(t')$$

where (a) follows because $\{Y(\tau) \leq 0\} \in \mathcal{F}(\tau) \subseteq \mathcal{F}(t')$ for all $\tau \in \{0, 1, \ldots, t'-1\}$ and $\{Y(t') > 0\} \in \mathcal{F}(t')$. It follows that $T$ is a stoping time.

2. **To show $\{\widetilde{Z}(t) \neq Z(t)\} \subseteq \bigcup_{\tau=0}^{t-1}\{Y(\tau) > 0\}, \forall t \geq 1$:** Fix $t = t' > 1$. Note that

$$\{\widetilde{Z}(t') \neq Z(t')\} \overset{(a)}{\subseteq} \{T < t'\} = \big\{ \inf\{t > 0 : Y(t) > 0\} < t' \big\}$$
$$\subseteq \bigcup_{\tau=0}^{t'-1} \{Y(\tau) > 0\}$$

where (a) follows by noting that if $T \geq t'$ then $\widetilde{Z}(t') = Z(t' \wedge T) = Z(t')$.

3. **To show $\widetilde{Z}(t)$ is a supermartingale with differences bounded by $c$:** Since random variable $T$ is proven to be a stoping time, $\widetilde{Z}(t) = Z(t \wedge T)$ is a supermartingale by Theorem 5. It remains to show $|\widetilde{Z}(t+1) - \widetilde{Z}(t)| \leq c, \forall t \geq 0$. Fix integer $t = t' \geq 0$. Note that

$$|\widetilde{Z}(t'+1) - \widetilde{Z}(t')|$$
$$= |Z(T \wedge (t'+1)) - Z(T \wedge t')|$$
$$= |\mathbf{1}_{\{T \geq t'+1\}}[Z(T \wedge (t'+1)) - Z(T \wedge t')] + \mathbf{1}_{\{T \leq t'\}}[Z(T \wedge (t'+1)) - Z(T \wedge t')]|$$
$$= |\mathbf{1}_{\{T \geq t'+1\}}[Z(t'+1) - Z(t')] + \mathbf{1}_{\{T \leq t'\}}[Z(T) - Z(T)]|$$
$$= \mathbf{1}_{\{T \geq t'+1\}}|Z(t'+1) - Z(t')|$$

Now consider $T \leq t'$ and $T \geq t'+1$ separately.

   - In the case when $T \leq t'$, it is straightforward that $|\widetilde{Z}(t'+1) - \widetilde{Z}(t')| = \mathbf{1}_{\{T \geq t'+1\}}|Z(t'+1) - Z(t')| = 0 \leq c$.
   - Consider the case when $T \geq t'+1$. By the definition of $T$, we know that $\{T \geq t'+1\} = \big\{ \inf\{t \geq 0 : Y(t) > 0\} \geq t'+1 \big\} \subseteq \bigcap_{\tau=0}^{t'}\{Y(\tau) \leq 0\} \subseteq \bigcap_{\tau=0}^{t'}\{|Z(\tau+1) - Z(\tau)| \leq c\}$, where the last inclusion follows from the fact that $\{|Z(\tau+1) - Z(\tau)| > c\} \subseteq \{Y(\tau) > 0\}$. That is, when $T \geq t'+1$, we must have $|Z(\tau+1) - Z(\tau)| \leq c$ for all $\tau \in \{1, \ldots, t'\}$, which further implies that $|Z(t'+1) - Z(t')| \leq c$. Thus, when $T \geq t'+1$, $|\widetilde{Z}(t'+1) - \widetilde{Z}(t')| = \mathbf{1}_{\{T \geq t'+1\}}|Z(t'+1) - Z(t')| \leq c$.

Combining two cases together proves $|\widetilde{Z}(t'+1) - \widetilde{Z}(t')| \leq c$.

Since $\widetilde{Z}(t)$ is a supermartingale with bounded differences $c$ and $\widetilde{Z}(0) = Z(0) = 0$, by the conventional Hoeffding-Azuma inequality, for any $z > 0$, we have

$$\Pr(\widetilde{Z}(t) \geq z) \leq e^{-z^2/(2tc^2)} \tag{30}$$

Finally, we have

$$
\begin{aligned}
\Pr(Z(t) \geq z) =& \Pr(\widetilde{Z}(t) = Z(t), Z(t) \geq z) + \Pr(\widetilde{Z}(t) \neq Z(t), Z(t) \geq z) \\
\leq& \Pr(\widetilde{Z}(t) \geq z) + \Pr(\widetilde{Z}(t) \neq Z(t)) \\
\overset{(a)}{\leq}& e^{-z^2/(2tc^2)} + \Pr(\bigcup_{\tau=0}^{t-1} Y(\tau) > 0) \\
\overset{(b)}{\leq}& e^{-z^2/(2tc^2)} + \sum_{\tau=0}^{t-1} p(\tau)
\end{aligned}
$$

where (a) follows from equation (30) and the second bullet in the above; and (b) follows from the union bound and the hypothesis that $\Pr(Y(\tau) > 0) \leq p(\tau), \forall \tau$.

### 7.9 Proof of Theorem 4

Define $Z(0) = 0$ and $Z(t) = \sum_{\tau=1}^{t} \sum_{k=1}^{m} Q_k(\tau) g_k^\tau(\mathbf{x}^*)$. Recall $\mathcal{W}(0) = \{\emptyset, \Omega\}$ and $\mathcal{W}(t) = \sigma(\omega(1), \ldots, \omega(t)), \forall t \geq 1$. The next lemma shows that for any $c > 0$, $Z(t)$ satisfies Lemma 9 with $\mathcal{F}(t) = \mathcal{W}(t)$ and $Y(t) = \|\mathbf{Q}(t+1)\| - \frac{c}{G}$.

**Lemma 13.** *Let $\mathbf{x}^* \in \mathcal{X}_0$ be any fixed solution that satisfies $\tilde{\mathbf{g}}(\mathbf{x}^*) \leq \mathbf{0}$, e.g., $\mathbf{x}^* = \operatorname{argmin}_{\mathbf{x} \in \mathcal{X}} \sum_{t=1}^{T} f^t(\mathbf{x})$. Let $c > 0$ be arbitrary. Under Algorithm 1, if we define $Z(0) = 0$ and $Z(t) = \sum_{\tau=1}^{t} \sum_{k=1}^{m} Q_k(\tau) g_k^\tau(\mathbf{x}^*), \forall t \geq 1$, then $\{Z(t), t \geq 0\}$ is a supermartingale adapted to filtration $\{\mathcal{W}(t), t \geq 0\}$ such that*

$$\{|Z(t+1) - Z(t)| > c\} \subseteq \{Y(t) > 0\}, \forall t \geq 0$$

*where $Y(t) = \|\mathbf{Q}(t+1)\| - \frac{c}{G}$ is a random variable adapted to $\mathcal{W}(t)$. (Note that $G$ is a constant defined in Assumption 1.)*

*Proof.* It is easy to say $\{Z(t), t \geq 0\}$ is adapted $\{\mathcal{W}(t), t \geq 0\}$. It remains to show $\{Z(t), t \geq 0\}$ is a supermartingale. Note that $Z(t+1) = Z(t) + \sum_{k=1}^{m} Q_k(t+1) g_k^{t+1}(\mathbf{x}^*)$ and

$$
\begin{aligned}
\mathbb{E}[Z(t+1)|\mathcal{W}(t)] =& \mathbb{E}[Z(t) + \sum_{k=1}^{m} Q_k(t+1) g_k^{t+1}(\mathbf{x}^*)|\mathcal{W}(t)] \\
\overset{(a)}{=}& Z(t) + \sum_{k=1}^{m} Q_k(t+1) \mathbb{E}[g_k^{t+1}(\mathbf{x}^*)] \\
\overset{(b)}{\leq}& Z(t)
\end{aligned}
$$

where (a) follows from the fact that $Z(t) \in \mathcal{W}(t)$, $\mathbf{Q}(t+1) \in \mathcal{W}(t)$ and $\mathbf{g}^{t+1}(\mathbf{x}^*)$ is independent of $\mathcal{W}(t)$; and (b) follows from $\mathbb{E}[g_k^{t+1}(\mathbf{x}^*)] = \tilde{g}_k(\mathbf{x}^*) \leq 0$ which further follows from $\omega(t)$ are i.i.d. samples. Thus, $\{Z(t), t \geq 0\}$ is a supermartingale.

We further note that

$$|Z(t+1) - Z(t)| = |\sum_{k=1}^{m} Q_k(t+1) g_k^{t+1}(\mathbf{x}^*)| \overset{(a)}{\leq} \|\mathbf{Q}(t+1)\| G$$

where (a) follows from the Cauchy-Schwarz inequality and the assumption that $\|\mathbf{g}^t(\mathbf{x}^*)\| \leq G$.

This implies that if $|Z(t+1) - Z(t)| > c$, then $\|\mathbf{Q}(t)\| > \frac{c}{G}$. Thus, $\{|Z(t+1) - Z(t)| > c\} \subseteq \{\|\mathbf{Q}(t+1)\| > \frac{c}{G}\}$. Since $\mathbf{Q}(t+1)$ is adapted to $\mathcal{W}(t)$, it follows that $Y(t) = \|\mathbf{Q}(t+1)\| - \frac{c}{G}$ is a random variable adapted to $\mathcal{W}(t)$. □

By Lemma 13, $Z(t)$ satisfies Lemma 9. Fix $T \geq 1$, Lemma 9 implies that

$$\Pr(\sum_{t=1}^{T}\sum_{k=1}^{m} Q_k(t)g_k^t(\mathbf{x}^*) \geq \gamma) \leq \underbrace{e^{-\gamma^2/(2Tc^2)}}_{(I)} + \underbrace{\sum_{t=0}^{T-1} \Pr(\|\mathbf{Q}(t+1)\| > \frac{c}{G})}_{(II)} \quad (31)$$

Fix $0 < \lambda < 1$. In the following, we shall choose $\gamma$ and $c$ such that both term (I) and term (II) in (31) are no larger than $\frac{\lambda}{2}$.

Recall that by Lemma 7, random process $\widetilde{Z}(t) = \|\mathbf{Q}(t)\|$ satisfies the conditions in Lemma 5 with $\delta_{\max} = G + \sqrt{m}D_2 R$, $\zeta = \frac{\epsilon}{2}$ and

$$\theta = \frac{\epsilon}{2}t_0 + (G + \sqrt{m}D_2 R)t_0 + \frac{2\alpha R^2}{t_0\epsilon} + \frac{2VD_1 R + [G + \sqrt{m}D_2 R]^2}{\epsilon}.$$

To guarantee term (II) is no lareger than $\frac{\lambda}{2}$, it suffices to choose $c$ such that

$$\Pr(\|\mathbf{Q}(t)\| > \frac{c}{G}) \leq \frac{\lambda}{2T}, \forall t \in \{1, 2, \ldots, T\}$$

By part (2) of Lemma 5 (with $\mu = \frac{\lambda}{2T}$), the above inequality holds if we choose $c = t_0\frac{\epsilon}{2}G + 2t_0(G + \sqrt{m}D_2 R)G + \frac{2\alpha R^2}{t_0\epsilon}G + \frac{2VD_1 R + [G + \sqrt{m}D_2 R]^2}{\epsilon}G + t_0\frac{8[G + \sqrt{m}D_2 R]^2}{\epsilon} \log[\frac{32[G + \sqrt{m}D_2 R]^2}{\epsilon^2}]G + t_0\frac{8[G + \sqrt{m}D_2 R]^2}{\epsilon} \log(\frac{2T}{\lambda})G$ where $t_0 > 0$ is an arbitrary integer.

Once $c$ is chosen, we further need to choose $\gamma$ such that term (I) in (31) is $\frac{\lambda}{2}$. It follows that if $\gamma = \sqrt{2T} \log^{0.5}(\frac{2}{\lambda})c = \sqrt{2T} \log^{0.5}(\frac{2}{\lambda})[\frac{\epsilon}{2}t_0 G + 2t_0(G + \sqrt{m}D_2 R)G + \frac{2\alpha R^2}{t_0\epsilon}G + \frac{2VD_1 R + [G + \sqrt{m}D_2 R]^2}{\epsilon}G + t_0\frac{8[G + \sqrt{m}D_2 R]^2}{\epsilon} \log[\frac{32[G + \sqrt{m}D_2 R]^2}{\epsilon^2}]G + t_0\frac{8[G + \sqrt{m}D_2 R]^2}{\epsilon} \log(\frac{2T}{\lambda})G]$, then the term (I) is equal to $\frac{\lambda}{2}$.

Thus, we have

$$\Pr(\sum_{t=1}^{T}\sum_{k=1}^{m} Q_k(t)g_k^t(\mathbf{x}^*) \geq \gamma) \leq \lambda,$$

which further implies,

$$\Pr(\sum_{t=1}^{T}\sum_{k=1}^{m} Q_k(t)g_k^t(\mathbf{x}^*) \leq \gamma) \geq 1 - \lambda. \quad (32)$$

Note that if we take $t_0 = \lceil\sqrt{T}\rceil$, $V = \sqrt{T}$ and $\alpha = T$, then $\gamma = O\big(T\log(T)\log^{0.5}(\frac{1}{\lambda})\big) + O\big(T\log^{1.5}(\frac{1}{\lambda})\big) = O\big(T\log(T)\log^{1.5}(\frac{1}{\lambda})\big)$.

By Lemma 8 (with $\mathbf{z} = \mathbf{x}^*$, $V = \sqrt{T}$ and $\alpha = T$), we have

$$\sum_{t=1}^{T} f^t(\mathbf{x}(t)) \leq \sum_{t=1}^{T} f^t(\mathbf{x}^*) + \sqrt{T}R^2 + \frac{D_1^2}{4}\sqrt{T} + \frac{1}{2}[G + \sqrt{m}D_2 R]^2\sqrt{T} + \frac{1}{\sqrt{T}}\sum_{t=1}^{T}\big[\sum_{k=1}^{m} Q_k(t)g_k^t(\mathbf{x}^*)\big] \quad (33)$$

Substituting (32) into (33) yields

$$\Pr\Big(\sum_{t=1}^{T} f^t(\mathbf{x}(t)) \leq \sum_{t=1}^{T} f^t(\mathbf{x}^*) + O\big(\sqrt{T}\log(T)\log^{1.5}(\frac{1}{\lambda})\big)\Big) \geq 1 - \lambda.$$

## 7.10 More Experiment Details

In the experiment, we assume the job arrivals $\omega(t)$ are Poisson distributed with mean 1000 jobs/slot. For simplicity, assume each server is restricted to choose power $x_i(t) \in [0, 30]$ at each round and

the service rate satisfies $h_i(x_i(t)) = 4\log(1 + 4x_i(t))$. (Note that our algorithm can easily deal with general concave functions $h_i(\cdot)$ and each server in general can have different $h_i(\cdot)$ functions.) The simulation duration is 2160 slots (corresponding to 10 days).

The three baselines are further elaborated as below:

- Best fixed decision in hindsight: Assume all the electricity price traces and the job arrival distribution are known beforehand. The decision maker chooses a fixed power decision vector $\mathbf{p}^*$ that is optimal based on data in 2160 slots.

- React algorithm: This algorithm is developed in [8]. The algorithm reacts to the current traffic and splits the load evenly among each server to support the arrivals. Since instantaneous job arrivals is unknown at the current slot, we use the average of job arrivals over the most recent 5 slots as an estimate. Since this algorithm is designed to meet the time varying job arrivals but is unaware of electricity variations, its electricity cost is high as observed in our simulation results.

- Low-power algorithm: This algorithm is adapted from [22] and always schedule jobs to servers in the zones with the lowest electricity price. Since instantaneous electricity prices are unknown at the current slot, we use the average of electricity prices over the most recent 5 slots at each server as an estimate. Recall that each server has a finite service capacity ($x_i(t) \in [0, 30]$), this algorithm is not guaranteed to serve all job arrivals. Thus, the number of unserved jobs can eventually pile up.