[Reviews · NeurIPS 2017]

Reviewer 1



This paper considers online convex optimization with time varying constraints. The goal is to minimize the regret similar to standard online convex optimization setting, and at the same time keep the average violation of stochastic constraints well-bounded. The authors address the problem in both fixed and stochastic settings and under mild conditions on the constraints, obtain the optimal \sqrt{T} log T bounds on both regret and violation of constraints that improves upon the best known T^{3/4} bound on the violation of constraints. The authors conducted preliminary experiments on a stochastic job scheduler that complements their theoretical results. The problem being dealt with is interesting and has proper motivation, though previous papers have dealt it with in the past, in one way or another, but this paper has shown a significant improvement over existing bounds on the regret and violation of constraints. The presentation of the paper was mostly clear. The claimed contributions are discussed in the light of existing results and the paper does survey related work appropriately. Regarding the quality of the writing, the paper is reasonably well written, the structure and language are good. The paper is technically sound and the proofs seem to be correct as far as I checked (I did not have enough time to carefully check the Appendix and just did few sanity checks, but definitely will take a careful look in rebuttal period). Overall, the problem dealt with consists of a new view to the online linear optimization with constraints that requires a new type of analysis. The exact setup of the feedback lacks motivation, but overall the idea of an analysis aimed to have long-term violation of constraints instead of expensive (and sometime impossible) projections in online learning is sufficiently interesting and motivated.

Reviewer 2



The paper considers online convex optimization with constraints revealed in an online manner. In an attempt to circumvent a linear regret lower bound by Mannor et al [17] for adaptively chosen constraints, the paper deals with the setting where constraints are themselves generated stochastically. As a side effect, superior results are obtained for related problems such as OCO with long-term constraints. The paper does a nice job of introducing previous work and putting the contribution in perspective. The main algorithm of the paper is a first order online algorithm that performs an optimization step using the instantaneous penalty and constraint functions. The formulation is similar to the one that leads, for example, to the classical Zinkevich's OGD update rule except for a term that involves the new constraint functions introduced at that time step. The formulation also uses an adaptive time varying regularization parameter (called a "virtual queue" in the paper) for the constraint terms in the formulation. Indeed, as Lemma 1 indicates, the solving the formulation is equivalent to a projected gradient descent (PGD)-style step, with the notion of "gradient" now including the constraint function subgradients as well. The virtual queue values are updated to remain lower bounds on the total constraint violation incurred till now. In terms of theoretical analysis, the arguments relating regret to drift etc seem routine. The most novel portion of the analysis seems to be the drift analysis and its adaptation to the ||Q(t)|| terms (Lemma 7). The paper reports one experimental use of the algorithm on a distributed job scheduling problem. The proposed algorithm is able to offer comparable throughput to a previously proposed approach called REACT while lowering costs by around 10%. - It would be nice if the paper could clearly point out the challenges in extending the PGD framework to incorporate time varying constraints. It seems except for the drift analysis, the rest of the arguments are routine after proper choices have been made for virtual queue values etc. - Please give some intuition on how was the form of the virtual queue values (equation 3) motivated. It seems to be a lower bound on the cumulative violation (or "backlog") but some intuition would much help. - Do the results presented in the paper extend to online mirrored descent? I do not see any obvious hurdles but some comments would be nice. - Experimental work is a bit sparse.

Reviewer 3



***Summary*** The paper proposes an algorithm to address a mixed online/stochastic setting where the objective function is a sequence of arbitrary convex functions (with bounded subgradients) under stochastic convex constraints (also with bounded subgradients). A static regret analysis (both in expectation and high-probability) is carried out. A real-world experiment about the allocation of jobs across servers (so as to minimize electricity cost) illustrates the benefit of the proposed method. Assessment: + Overall good & clear structure and presentation + Technically sound and appears as novel + Good regret guarantees - Confusing connection with "Deterministic constrained convex optimization" - Some confusion with respect to unreferenced previous work [A] - More details would be needed in the experimental section More details can be found in the next paragraph. ***Further comments*** -In Sec. 2.2, the rationale of the algorithm is described. In particular, x(t+1) is chosen to minimize the "drift-plus-penalty" expression. Could some intuition be given to explain how the importance of the drift and penalty are traded off (e.g., why is a simple sum, without a tuning weight, sufficient?). -Lemma 5: I do not really have the expertise to assess whether Lemma 5 corresponds to "a new drift analysis lemma for stochastic process". In particular, I am not sufficiently versed in the stochastic process literature. -Could some improvements be obtained by having \alpha and V dependent on t? -For Lemma 9, it seems to me that the authors could reuse the results from Proposition 34 in [B] -Comparison with "OCO with long term constraints": It appears that [A] (not referenced) already provide O(sqrt(T)) and O(sqrt(T)) guarantees, using similar algorithmic technique. This related work should be discussed. -Comparison with "Stochastic constrained convex optimization": Is [16] the only relevant reference here? -Confusion comparison with "Deterministic constrained convex optimization": In the deterministic constrained setting, we would expected the optimization algorithm to output a feasible solution; why is it acceptable (and why does it make sense) to have a non-feasible solution here? (i.e., constraint violation in O(1/sqrt(T))) -Experiments: - More details about the baselines and the implementation (to make things reproducible, e.g., starting points) should appear in the core article - Is the set \mathcal{X_0} = [x_1^min, x_1^max] \times ... \times [x_100^min, x_100^max]? If this is the case, it means that problem (2) has to be solved with box constraints. More details would be in order. - Log-scale for unserved jobs (Figure d) may be clearer - Bigger figures would improve the clarity as well - An assessment of the variability of the results is missing to decide on the significance of the conclusions (e.g., repetitions to display standard errors and means). - To gain some space for the experiments, Sec. 4 could be reduced and further relegated to the supplementary material. -Could the analysis extended to the dynamic regret setting? ***Minor*** -line 52: typo, min -> argmin. I may have missed them in the paper, but under which assumptions does the argmin of the problem reduce to the single x^*? -line 204 (a): isn't it an inequality instead of an equality? [A] Yu, H. & Neely, M. J. A Low Complexity Algorithm with O(sqrt(T)) Regret and Constraint Violations for Online Convex Optimization with Long Term Constraints preprint arXiv:1604.02218, 2016 [B] Tao, T.; Vu, V. Random matrices: universality of local spectral statistics of non-Hermitian matrices The Annals of Probability, 2015, 43, 782-874 ================== post-rebuttal comments ================== I thank the authors for their rebuttal (discussion about [A] was ignored). I have gone through the other reviews and I maintain my score. I would like to emphasize though, that the authors should clarify the discussion about the stochastic/deterministic constrained convex opt. case (as answered in the rebuttal).